https://doi.org/10.1038/s41467-022-30766-x | OPEN

# Boosting the performance of single-atom catalysts via external electric field polarization

Yanghang Pan[1,7], Xinzhu Wang[1,2,7], Weiyang Zhang[3], Lingyu Tang[1], Zhangyan Mu[1], Cheng Liu[1], Bailin Tian[1], Muchun Fei[1], Yamei Sun[1], Huanhuan Su[4], Libo Gao [4], Peng Wang [3,5], Xiangfeng Duan [6✉], Jing Ma [1,2✉] & Mengning Ding [1✉]

Single-atom catalysts represent a unique catalytic system with high atomic utilization and tunable reaction pathway. Despite current successes in their optimization and tailoring through structural and synthetic innovations, there is a lack of dynamic modulation approach for the single-atom catalysis. Inspired by the electrostatic interaction within specific natural enzymes, here we show the performance of model single-atom catalysts anchored on two-dimensional atomic crystals can be systematically and efficiently tuned by oriented external electric fields. Superior electrocatalytic performance have been achieved in single-atom catalysts under electrostatic modulations. Theoretical investigations suggest a universal "onsite electrostatic polarization" mechanism, in which electrostatic fields significantly polarize charge distributions at the single-atom sites and alter the kinetics of the rate determining steps, leading to boosted reaction performances. Such field-induced on-site polarization offers a unique strategy for simulating the catalytic processes in natural enzyme systems with quantitative, precise and dynamic external electric fields.

[1] Key Laboratory of Mesoscopic Chemistry, School of Chemistry and Chemical Engineering, Nanjing University, 210023 Nanjing, Jiangsu, China. [2] Jiangsu Key Laboratory of Advanced Organic Materials, School of Chemistry and Chemical Engineering, Nanjing University, 210023 Nanjing, Jiangsu, China. [3] National Laboratory of Solid State Microstructures, Jiangsu Key Laboratory of Artificial Functional Materials, College of Engineering and Applied Sciences, Collaborative Innovation Center of Advanced Microstructures, Nanjing University, 210093 Nanjing, Jiangsu, China. [4] National Laboratory of Solid State Microstructures, School of Physics, Collaborative Innovation Center of Advanced Microstructures, Nanjing University, 210093 Nanjing, Jiangsu, China. [5] Department of Physics, University of Warwick, Coventry CV4 7AL, UK. [6] Department of Chemistry and Biochemistry, University of California, Los Angeles, CA 90095, USA. [7]These authors contributed equally: Yanghang Pan, Xinzhu Wang. ✉email: xduan@chem.ucla.edu; majing@nju.edu.cn; mding@nju.edu.cn

The electric field plays a key role in enzyme catalysis, with a significant contribution to the local electrostatics and protein dynamics that allowed slow and difficult chemical reactions to occur rapidly at the active centers[1–3]. To mimic such field effect in natural systems, oriented external electric fields (OEEFs) have been proposed[4–8] for modulating catalytic activity and selectivity in single-molecule reactions[9–11]. Specifically, electric fields can influence electronic interactions (vibrational frequencies, adsorption structures, and orbital orientation) in adsorbate/catalytic material interfaces and alter thermodynamic and kinetic properties of elementary reactions[12]. Moreover, with the rapid development of micro/nano-device technology, the electrical transport spectroscopy (ETS), and similar in-device electrocatalysis have emerged as a versatile platform for quantitative analysis at the micro/nanoscale[13–15], where the OEEFs can be easily implemented, providing precisely controlled states for the nanoscale catalysts, particularly the two-dimensional (2D) atomic crystals[16–20].

Single-atom catalysts (SACs) have arguably become one of the most active frontiers in heterogeneous catalysis for its high atom utilization efficiency[21,22], featuring highly tunable active centers with well-defined coordination geometric/electronic structures similar to typical homogeneous catalysts or nanozymes[23–26]. In this regard, SACs offer an attractive model catalyst for mimicking the natural biocatalytic systems and exploring the corresponding catalytic pathways[27–29]. Despite enormous research focusing on the compositional design and synthesis of SACs with different coordination environments or local electronic structures[30–34], there is a lack of an effective approach to dynamically modulate the activity of SACs through external intervention during the catalytic process. Meanwhile, it has been suggested that the protrusion-shaped SAs can produce internal local electric fields (tip effect) in specific directions (determined by the fixed electronic structure of SACs) and may greatly influence the reactant activation[35,36] and ion distribution in electrolyte[37]. To this end, it is of great significance to explore the electrostatic effect on the system of SACs in a controlled and quantitative manner, which may offer a precise and universal model for the mechanistic evaluations, and the development of high-performance catalysts.

Here we demonstrate the quantitative manipulation of local catalytic sites of single-atom catalyst anchored on 2D atomic crystals via a precisely controlled external electric field. A series of model SACs supported on different 2D substrates have been systematically tested in the microcell, and their electrocatalytic performance can be dynamically tailored by OEEFs. Specifically, the Pt SAs on n-type $MoS_2$ (Pt SAs-$MoS_2$) demonstrate significant improvement in HER performance under positive OEEFs ($\eta_{HER}$ of 20 mV @10 mA $cm^{-2}$, Tafel slope of 51 mV $dec^{-1}$, at $V_g$ of +40 V), and the Co SAs on p-type $WSe_2$ (Co SAs-$WSe_2$) achieve a dramatically promoted activity for OER under negative OEEFs ($\eta_{OER}$ of 139 mV at 10 mA $cm^{-2}$, Tafel slope of 64 mV $dec^{-1}$, at $V_g$ of −40 V). In situ investigations, electrokinetic studies and density functional theory (DFT) calculations reveal an "onsite electrostatic polarization" mechanism, where the vertical electric field effectively modulates the charge distribution at the SA sites, polarizing the frontier orbitals of metal atoms/adsorbates/intermediates and significantly modifying the activation energies and/or reaction pathways during the catalytic processes. This unique onsite modulation mode can dynamically promote the electrocatalytic performance of SACs, leading to record-high activities, and serves as a highly tunable physical model for the enzyme-mimic artificial modulation and in-depth understanding of general heterogeneous/electro-catalysis.

## Results and discussion
### Micro-device construction and characterization of model SACs on 2D atomic crystals. Electrolyte gating occurs spontaneously to

make semiconductors more conductive during the electrochemical process[38]. As shown in Fig. 1a, Pt SAs was first deposited on a prefabricated micro-device of n-type $MoS_2$ through UV light-assisted reduction. For the systematic study, a series of model SACs (combination of Pt/Co/Pd and n-type $MoS_2$/p-type $WSe_2$/metallic-graphene) were also prepared via similar approaches. The detailed synthesis steps of 2D SACs and fabrication of micro-device had been shown in "Methods" and Supplementary Information. Aberration-corrected high-angle annular dark-field scanning transmission electron microscopy (HAADF-STEM), which has been widely adopted in microstructure characterization[39], demonstrated that Pt SAs were successfully introduced into the 2H-$MoS_2$ lattice without disturbing its crystalline structure (Fig. 1b–e and Supplementary Figs. 1–4). The Mo:S atomic ratios (Supplementary Table 2) are determined to be 1.00:1.98 and 1.00:1.96 in bulk $MoS_2$ and Pt SAs-$MoS_2$, respectively, further confirming that the near-perfect lattice (few defects) was maintained. The area density of Pt SAs on mechanically exfoliated $MoS_2$ is determined to be 0.42 atom·$nm^{-2}$ (~4.84 wt%) via statistical analysis of the STEM images (Supplementary Table 1). Raman and photoluminescence spectroscopy (Fig. 1f and Supplementary Fig. 8) revealed the doping effect of Pt SAs on 2H-$MoS_2$ nanosheets[40–43], which also leads to a more positive threshold voltage in the electrical FET transfer curves (Fig. 1g and Supplementary Fig. 7) of n-type $MoS_2$ micro-device modified by Pt SAs[44–46].

The chemical configuration and local coordination environment of Pt SAs-$MoS_2$ were then investigated through the combination of X-ray photoelectron spectroscopy (XPS) and X-ray absorption spectroscopy (XAS). Compared to that of commercial Pt/C and Pt NPs-$MoS_2$ located at around 71.0 eV (for $Pt^0$), the Pt $4f_{7/2}$ XPS peak (Fig. 1h) of Pt SAs-$MoS_2$ significantly moved towards higher binding energies (approximately 72.5-73.0 eV), with obvious alteration in peak shape. Moreover, the results of peak deconvolution further demonstrated that the Pt species in Pt SAs-$MoS_2$ were partially oxidized to the valence state of 0-+4 ($Pt^{\delta+}$), mainly composed of $Pt^{2+}$ and $Pt^{4+}$ (72.3 and 73.4 eV, Supplementary Fig. 11), which should be attributed to the electronic interactions between Pt SAs and $MoS_2$ nanosheets[47]. Moreover, in X-ray absorption near-edge spectroscopy (XANES) the Pt $L_3$-edge analysis (Fig. 1i) showed that the white line intensity for Pt SAs-$MoS_2$ was higher than that for Pt foil, indicating the oxidized Pt species were not in the metallic state[47–49]. As shown by EXAFS in R space (Fig. 1j), Pt SAs-$MoS_2$ exhibited a peak at approximately 2.3 Å from the Pt-S shell with a coordination number of 4.0 (Supplementary Table 3). Besides, no typical peaks for Pt-Pt bond at longer distances (>2.7 Å) were detected, revealing the atomic dispersion of Pt atoms on $MoS_2$ nanosheets. More importantly, the theoretical distance of Pt-S bond (2.291 Å, Supplementary Table 11) from the optimized model of (M-top)-Pt SAs-$MoS_2$ (more details in Supplementary Fig. 88) was also consistent with the EXAFS results, further confirming the Mo-atop site of Pt SAs anchored on $MoS_2$.

### OEEF modulation in SA-catalyzed HER. On-chip electrochemical and in situ transport investigations were simultaneously conducted in the four-electrode microcell (Fig. 2a and Supplementary Figs. 23–25), and the vertical electric fields can be applied through back-gate electrode[50,51]. To set a background, modulation of electrochemical behaviors under electric fields was first investigated on the pristine $MoS_2$ with different thicknesses (Supplementary Fig. 27). As shown in Fig. 2b and Supplementary Figs. 28–30, the HER performance of n-type $MoS_2$ with less layers can be tailored more efficiently by OEEFs. This thickness-

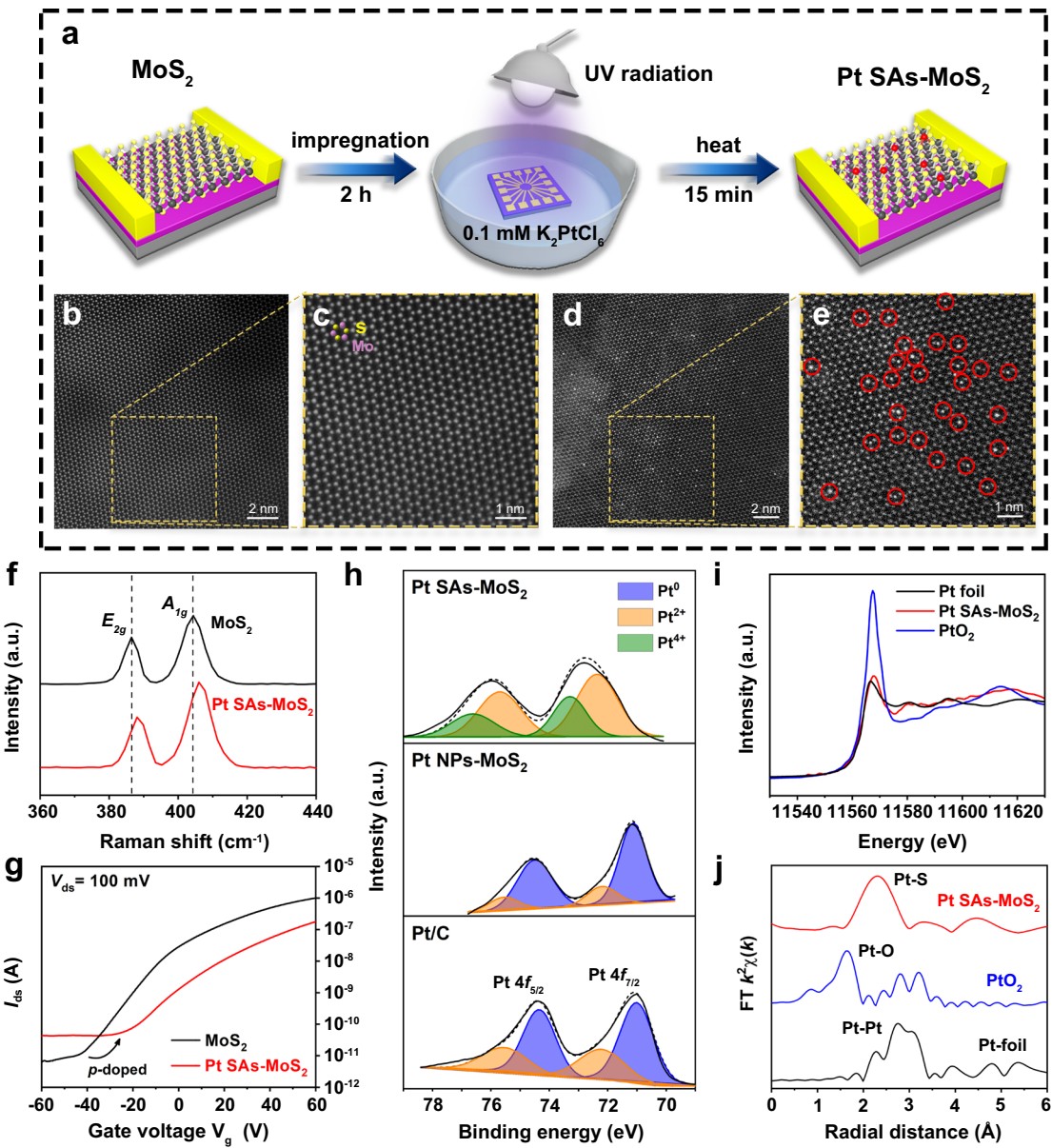

**Fig. 1 Synthesis and characterizations of Pt SAs-MoS₂. a** Proposed reaction mechanism for the preparation of Pt SAs-MoS₂ micro-device. **b, c** HAADF-STEM images of mechanically exfoliated monolayer MoS₂. Specifically, **c** is the magnified image of yellow dashed area in **b**. **d, e** HAADF-STEM images of Pt SAs-MoS₂. Specifically, **e** is the magnified image of yellow dashed area in **d**, with red circles highlighting the Pt SAs distributed in the 2H-MoS₂ lattice. **f** Raman spectroscopy of mechanically exfoliated monolayer MoS₂ (black) and Pt SAs-MoS₂ (red). **g** Transfer characteristics of FET devices fabricated from mechanically exfoliated monolayer MoS₂ (black) and Pt SAs-MoS₂ (red). **h** Peak deconvolution of Pt $4f$ XPS spectra for commercial Pt/C, Pt NPs-MoS₂, and Pt SAs-MoS₂. **i** Normalized Pt $L_3$-edge XANES spectra of Pt foil, Pt SAs-MoS₂, and PtO₂. **j** First shell fitting of EXAFS spectra for Pt foil, Pt SAs-MoS₂, and PtO₂.

selectivity on pristine 2D TMDs[20,38] originates from the Debye screening effect[52,53], which provides a background behavior of 2D support for this study. On this basis, HER tests (with in situ transport measurement of $I_{ds}$) were performed on Pt SAs-MoS₂ with few-layer MoS₂ (<5 nm) as the support. To quantify the modulation efficiency of OEEFs on electrocatalysis, we define a "tunable potential range ($\Delta\eta_E$)" (inset of Fig. 2b), with $V_g$ (275 nm SiO₂ as the dielectric layer) varying from −40 V to +40 V at 50 mA cm⁻². Figure 2c shows that Pt SAs-MoS₂ (red/blue solid lines) exhibits considerably higher HER performance over pristine MoS₂ (gray dashed lines), with a further and significant promotion ($\Delta\eta_E = 52$ mV) under vertically applied positive electric field. Meanwhile, Tafel slopes (Fig. 2d) of Pt SAs-MoS₂ also decline from 117 to 65 mV dec⁻¹ with $V_g$ applied from −60 V to +60 V, which is much smaller than the pristine 2D support without SA decorations (from 161 to 105 mV dec⁻¹, consistent with the pristine 2H-MoS₂ for HER[54–56]) and gradually approaches Pt-based electrocatalysts reported previously[57–59]. Note that negligible back-gate leakage current (Supplementary Figs. 86 and 87) was observed, indicating charge accumulated at the MoS₂–SiO₂ interface was kept stable during the top-gate electrochemical process, and no extra energy input is required by the introduction of an electrostatic field via the back-gate circuit. Such continuous promotion at the high-activity region demonstrates the potential of an externally applied electric field to further push the already effective SACs toward superior performance, presumably through alternation in catalytic kinetics.

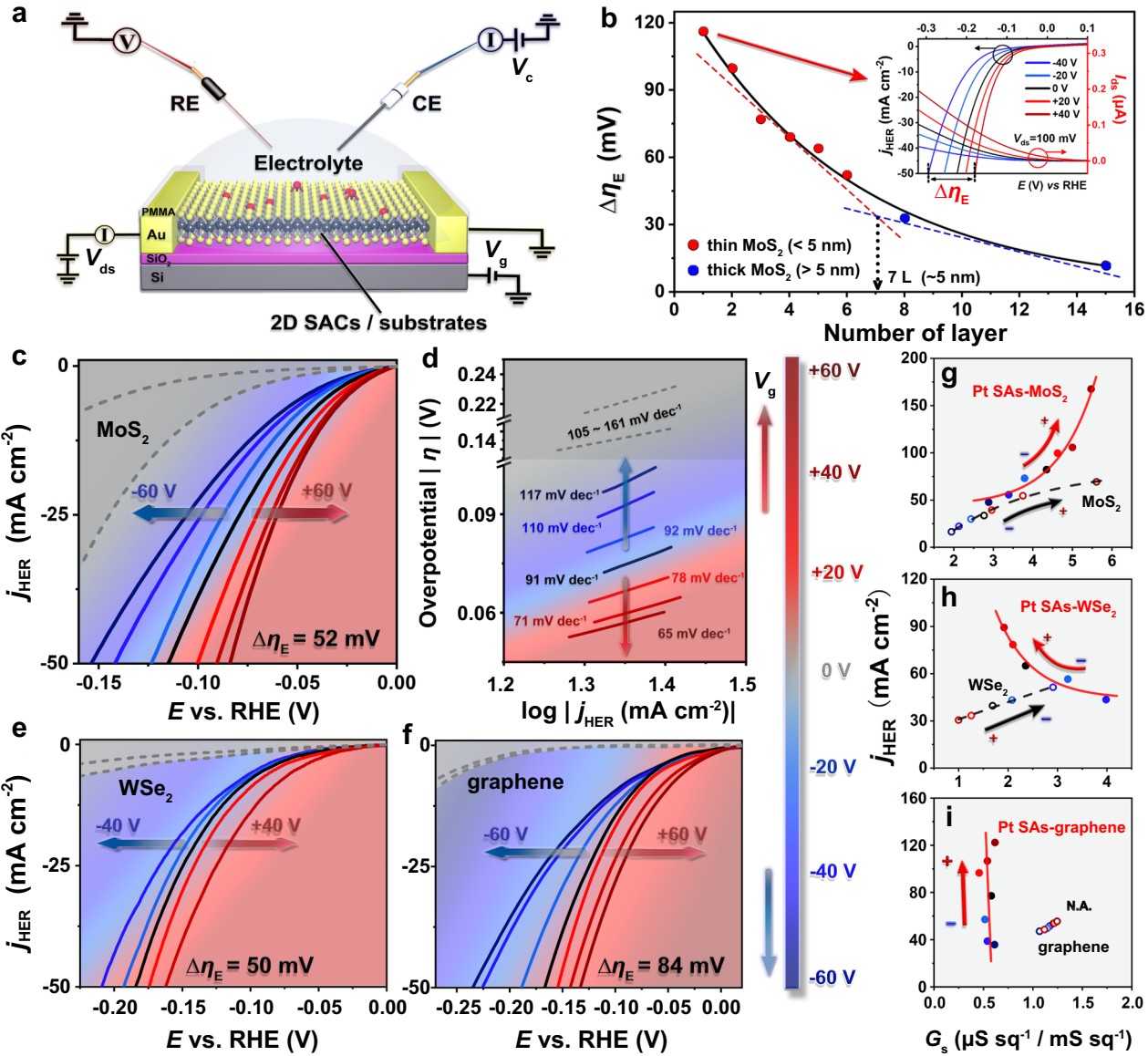

**Fig. 2 In situ electronic/electrochemical investigations on Pt-based SACs for HER under oriented external electric field (OEEF) regulation. a** Schematic diagram of an on-chip four-electrode microcell for electronic/electrochemical measurements, OEEFs were introduced via vertically applied back gate voltage ($V_g$) through a dielectric layer. **b** Layer dependence on the HER activity modulation (tunable potential range, $\Delta\eta_E$) in pristine 2D MoS$_2$ (n-type semiconductor). Inset: HER polarization curves and in situ drain-source current of monolayer MoS$_2$ under OEEF regulation, with $V_{ds} = 100$ mV and $V_g$ from $-40$ V to $+40$ V. **c, d** HER polarization curves (**c**) and Tafel curves (**d**) of Pt SAs-MoS$_2$ in response to OEEFs. **e, f** HER polarization curves of Pt SAs-WSe$_2$ (**e**) and Pt SAs-graphene (**f**) in response to OEEFs. The two gray dashed lines in **c**–**f** represent the tunable HER activity of pristine 2D substrates under OEEF regulation. The color bar in **c**–**f** shows the direction and scope of $V_g$. **g**–**i** Correlations between electrochemical current density ($j_{HER}$) and in situ sheet conductance ($G_s$) of Pt SAs-MoS$_2$ (**g**), Pt SAs-WSe$_2$ (**h**), Pt SAs-graphene (**i**), and pristine 2D substrates under OEEF regulation. The arrows in **g**–**i** indicate the OEEF-modulation directions of HER performance optimization. Note that in **g**–**i** the unit of abscissa is $\mu$S sq$^{-1}$ for Pt SAs-MoS$_2$ and Pt SAs-WSe$_2$, while mS sq$^{-1}$ for Pt SAs-graphene, and $j_{HER}$ in **g**–**i** were extracted at the potentials (vs. RHE) of: $-0.15$ V for Pt SAs-MoS$_2$, $-0.2$ V for MoS$_2$, $-0.2$ V for Pt SAs-WSe$_2$, $-0.4$ V for WSe$_2$, $-0.2$ V for Pt SAs-graphene, $-0.4$ V for graphene, to obtain values of similar level for better comparison.

To probe the mechanism of this electric-field modulation on SACs, we extracted in situ sheet conductance ($G_s$, derived from $I_{ds}$ at different OEEFs, Supplementary Fig. 33) in Pt SAs supported on n-type MoS$_2$ and studied its correlation with HER current density ($j_{HER}$), as shown in Fig. 2g. Similar ETS signals have been proved effective to identify the operando charge carrier states in pristine semiconductor catalysts[38,60], and are presumably informative for the more comprehensive single catalytic sites. First, a positive and close-to-linear $j_{HER}$-$G_s$ correlation (black dashed line in Fig. 2g) was established in pristine MoS$_2$. In this case, carrier (electron) injection into n-type semiconductors can lead to the

gradually declined overpotential originated from inner resistance (contact and/or sheet resistance of MoS$_2$), and the rise of their Fermi level reduces the corresponding energy barrier in HER (i.e., change of H adsorption ability), thus promoting the HER kinetics[17,18,61]. In contrast, Pt SAs-MoS$_2$ (red solid line in Fig. 2g) exhibited a distinct near-exponential $j_{HER}$-$G_s$ correlation, indicating a different modulation mechanism in single-atom electrocatalysis, where field-induced carrier injection (i.e., the gating effect) in 2D supports is not the major cause for HER modulation in SACs. Considering the structural and catalytic features of the SACs, we propose an alternative theory that the OEEF

modulation of the catalytic performance could originate its direct influence on the SA catalytic sites. To further verify this hypothesis, OEEF-induced HER modulation was investigated in Pt SAs anchored on p-type $WSe_2$ (Fig. 2h) and metallic-graphene (Fig. 2i), where electric gating will lead to different conductivity trends in the 2D supports. Indeed, pristine p-type $WSe_2$ shows an opposite direction in both $G_s$-$V_g$ and $j_{HER}$-$G_s$ relationships compared to n-type $MoS_2$ (note that $G_s$ increases at different $V_g$ directions for n- and p-type TMDs, as indicated by the arrows in Fig. 2g, h), as a result of the same field modulation mechanism through charge carrier injection. In contrast, a negative $j_{HER}$-$G_s$ correlation (red solid line in Fig. 2h) was observed in Pt SAs-$WSe_2$ (same as Pt SAs-$MoS_2$), which confirms that SACs follow an alternative OEEF modulation mechanism aside from the carrier injection in 2D supports. More obviously, negligible electric-field response ($\Delta\eta_E < 10$ mV, Supplementary Fig. 43) was observed in pristine graphene due to the relatively smaller modification in its already large carrier concentrations under different OEEFs, whereas the HER performance of Pt SAs-graphene was again significantly enhanced by positive OEEFs ($\Delta\eta_E = 84$ mV, Fig. 2f), with almost no fluctuation in $G_s$ (red solid line in Fig. 2i). Overall, it can be concluded that for HER performance under OEEFs, 2D supports follow the typical mechanism of carrier-concentration modulation in semiconductors that leads to different trends in n-type, p-type, and metallic supports, while the electrocatalytic performance promotion in Pt SAC systems was unanimously triggered by positive OEEFs regardless of the different behaviors in 2D supports. These results provide strong evidence that the OEEF modulation in SACs is originated from a distinctive mechanism, probably through the direct modification on the single-atom catalytic sites.

To rationalize the unique reaction pathway for the OEEF-induced catalytic modulation in Pt SACs, theoretical calculations were conducted to investigate their physicochemical characteristics under OEEFs. A 4×4 supercell of monolayer 2H-$MoS_2$ with 3.1% sulfur vacancies, which are usually regarded as the main active sites in the basal plane of 2H-$MoS_2$ for HER[62–64], was set up for HER studies (see more details in "Methods"). The hydrogen adsorption free energy ($\Delta G_H$), which is widely recognized as a descriptor for HER activity[65,66], was investigated by direct introduction of electric field gradients ($E_{field}$) into the supercell (Fig. 3a). For Pt SAs, the $M$-top site binding to S atom, which was confirmed by XAS results and STEM observations, was constructed as the optimal DFT model for subsequent theoretical analysis. As shown in Fig. 3b, $\Delta G_H$ of ($M$-top)-Pt SAs-$MoS_2$ becomes noticeably more negative (red solid line) when placed in a vertical electric field from −0.4 V/Å to +0.4 V/Å, whereas $\Delta G_H$ of 2H-$MoS_2$ slightly moves towards the zero point (black solid line) in response to $E_{field}$. Accordingly, the partial density of states (PDOS) projected onto the H 1s and the Pt $5d_{z^2}$ orbitals show consistent response to OEEFs, and the energy gap between the centers of H 1s-Pt $5d_{z^2}$ bonding states and antibonding states become smaller when applying positive OEEFs (Supplementary Fig. 92). In this case, less energy is required to form the hybridized orbitals[67–69] and thus higher *H adsorption energy on Pt SAs-$MoS_2$ (a more compact Pt-H bond) is observed. Solvation model of ($M$-top)-Pt SAs-$MoS_2$ was further constructed to investigate the possible effect from solvent (water) molecules at the electrochemical interface (Supplementary Fig. 90). The values of hydrogen adsorption energy in presence of a water molecule exhibited a negative correlation with the introduced electric field gradients, which is similar to the $\Delta G_H$-$E_{field}$ relationship observed in the gas-phase model (Supplementary Fig. 91). More importantly, the trend of OEEF-modulated charge distribution on the adsorbed *H ($Q_H$, Supplementary Table 15) is also close to the gas-phase model.

The declining of $\Delta G_H$ on Pt SAs-$MoS_2$ is actually moving away from the volcano apex (near the zero point) that is favorable for HER[70], thus inconsistent with the observed OEEF modulation on HER activities. These results indicate that the OEEF-modulation of HER in Pt SACs essentially went through a different mechanism, which is also indicated by comparing the Tafel characteristics observed in Pt SAs-$MoS_2$ and $MoS_2$ (Fig. 2d). In principle, a Tafel slope around 120 mV dec$^{-1}$ or higher suggests that the first H adsorption (the Volmer reaction) limits the overall reaction rate in HER[71]. Therefore, the mediocre values of the Tafel slope (over the whole range of $V_g$) observed in $MoS_2$ (105–161 mV dec$^{-1}$) signifies that the HER optimization should be attributed to an increase in carrier concentrations (rather than a slight change in $\Delta G_H$) under electric field regulation[17–19]. Distinctively, the smaller Tafel slope of Pt-$MoS_2$ (65–117 mV dec$^{-1}$) demonstrates that the response in HER activities to OEEFs in the single-atom system goes beyond the limitation of the Volmer step and presumably originates from the modulation on the overall kinetics. It is also interesting to notice that Pt SAs-$MoS_2$ exhibits a unique HER behavior compared to those of Pt NPs-$MoS_2$ (Supplementary Fig. 51) and Pt microelectrode (Supplementary Fig. 53), with lower onset potential but slightly larger Tafel slope (Supplementary Table 9). This can be rationalized by that Pt SAs-$MoS_2$ follows the Volmer-Heyrovsky mechanism[71] with a theoretical Tafel slope of 40 mV dec$^{-1}$ (rate-determining step is the Heyrovsky reaction), while other Pt species (with relative high Pt content) are usually limited by the Tafel reaction with theoretical Tafel slope of 30 mV dec$^{-1}$. On these bases, OEEFs presumably act on the Heyrovsky reaction to further influence the electro-kinetics.

The charge analysis (Supplementary Figs. 93 and 94 and Supplementary Table 14) of H adsorption on 2H-$MoS_2$ and Pt SAs-$MoS_2$ was then performed to reveal the in-depth electrocatalytic regulation mechanism induced by OEEFs. As is shown in Fig. 3c, the outward electron density around the first adsorbed *H site exhibits a significant enhancement under the positive $E_{field}$, with increased $Q_H$ (top) and less electron depletion (bottom). For comparison, injection of excess electrons equivalent to $V_g$ from −60 V to 60 V (Supplementary Table 16) was also performed, which generated similar results (Supplementary Figs. 96 and 97). Note that such modulation was based on charge "polarization" directly on the *H site, rather than through the modulation of carrier concentrations as in the electric field-induced charge injections, which explains the universal trend in $j_{HER}$-$V_g$ correlations observed in Pt SACs anchored on different types of 2D supports (p-, n-, and metallic), as indicated by red arrows in Fig. 2g–i. This electrostatic polarization certainly benefits the approaching of a second proton (H$^+$) to the Pt-H intermediate and thus significantly promotes the kinetics of the Heyrovsky reaction in HER, as summarized in Fig. 3d. Overall, positive OEEFs effectively lower the energy barrier through outward orbital polarization at the *H site to promote the Heyrovsky step, pushes the desorption of H$_2$ molecules originally restricted by the strong $\Delta G_H$, and thus breaking through the limiting volcano plots of HER to achieve a superior activity, as illustrated in Fig. 3e. This OEEFs-triggered mechanism appears specific in SAC systems (regardless of the modulation on carrier concentrations that existed in 2D substrates), and can be understood as "onsite electrostatic polarization" in single-atom electrocatalysis, boosting SACs kinetics via an artificial enzyme-mimicking approach to achieve superior performance.

## OEEF modulation of SACs for OER through "onsite electrostatic polarization". The "onsite electrostatic polarization" mechanism discovered in SACs was further extended to the

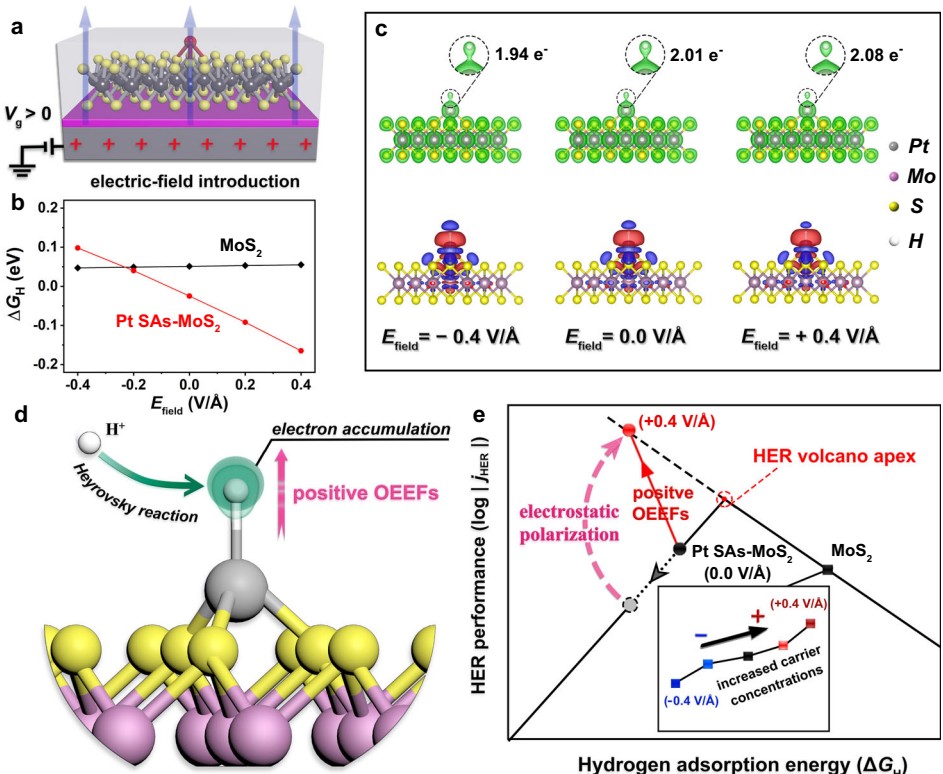

**Fig. 3 "Onsite electrostatic polarization" in the Pt SAs-MoS₂ catalyzed HER. a** Schematic diagram of the electric-field introduction into Pt SAs-MoS₂. **b** Hydrogen adsorption energy ($\Delta G_H$) in the models of 2H-MoS₂ (black) and (*M-top*)-Pt SAs-MoS₂ (red), with $E_{field}$ from −0.4 V/Å to +0.4 V/Å. **c** Charge analysis of (*M-top*)-Pt SAs-MoS₂ under OEEF regulation, where *H is adsorbed on Pt atom. Top: charge densities with interface values of 0.15 e⁻/Bohr³. The zoomed-in area shows the charge distribution at the first adsorbed *H site. Bottom: deformation charge densities with interface values of 0.001 e⁻/Bohr³. Red and blue areas denote electron accumulation and depletion, respectively. $E_{field}$ from left to right is −0.4 V/Å, 0, and +0.4 V/Å. **d** Schematic diagram of the "onsite electrostatic polarization" mechanism on Pt SAs-MoS₂ for HER. During the Heyrovsky reaction, H⁺ in electrolyte tends to move towards the first adsorbed *H site, where electron accumulation was triggered by positive OEEFs. **e** The positive-OEEF-induced breakthrough at HER volcano apex ($j_{HER}$-$\Delta G_H$) in Pt SAs-MoS₂. Inset: HER performance of pristine MoS₂ associated with the OEEF-regulated carrier concentrations (consistent with the $j_{HER}$-$G_s$ correlation observed in Fig. 2g), rather than the slight change in $\Delta G_H$ (black solid line in **b**).

oxidation reaction. Co SAs-WSe₂ was used as a model catalyst as it combines an OER favorable SAC and a p-type substrate. As confirmed by HAADF-STEM images (Supplementary Figs. 12–14), Co SAs were successfully introduced into the mechanically exfoliated WSe₂, with an area density of 0.41 atom·nm⁻² (~0.78 wt%, Supplementary Table 4). The Co 2p XPS signals (Supplementary Fig. 20) also indicated that the oxidized Co species (Co^{δ+}) in Co SAs-WSe₂ is different from Co nanoparticles. As expected, Co SAs-WSe₂ exhibits dramatically improved OER activity under the negative OEEF modulation. Both onset potential and Tafel slope of OER were optimized when negative $V_g$ was applied on Co SAs-WSe₂ (Fig. 4a, d), which was also observed in Co SAs-MoS₂ (Fig. 4b) and Co SAs-graphene (Fig. 4c). More importantly, unique $j_{OER}$-$G_s$ correlations (Fig. 4e–g) were observed in Co SACs, distinct from the close-to-linear behaviors in pristine 2D substrates. For all substrates (p-WSe₂, n-MoS₂, and metallic-graphene), the OEEF modulation directions were identical to the alteration of carrier concentrations in response to the applied $V_g$ (black arrows in Fig. 4e–g). As a result, p-type WSe₂ and n-type MoS₂ showed opposite modulation directions (indicated by black arrows in Fig. 4e, f). For all SACs, the trend of OER modulation is unanimously correlated to the OEEF direction regardless of the different trends in $G_s$ (red arrows in Fig. 4e–g), which signifies the presence of "onsite electrostatic polarization" in anodic electrocatalysis.

All possible configurations of Co atom anchored on 2H-WSe₂ were considered and three DFT models were constructed

(Supplementary Fig. 98). In line with the STEM observations, the two optimal models with lower formation energies (*M-top* and *X-sub*) were ultimately applied for subsequent theoretical analysis by introducing different $E_{field}$, with successive *OH, *O, and *OOH adsorptions. Figure 4h shows the adsorption free energies ($\Delta G_{OH}$, $\Delta G_O$, $\Delta G_{OOH}$) of three oxygen intermediates along the overall OER pathway on (*M-top*)-Co SAs-WSe₂. Specifically, the adsorption free energies can be effectively tailored by the external electric field, accompanied by obvious changes in free energy differences ($\Delta G_1$, $\Delta G_2$, $\Delta G_3$, $\Delta G_4$). And the maximum of free energy difference ($\Delta G_3$, see "Methods" and Supplementary Tables 19 and 20) indicated that the conversion from *O to *OOH was the potential-determining step (pds) in Co SAs-WSe₂ for OER. As shown in Fig. 4i, $\Delta G_O$ of (*M-top*)-Co SAs-WSe₂ moves towards the zero point when $E_{field}$ goes negatively (red solid line), whereas WSe₂ has no obvious change in $\Delta G_O$ (black solid line). Meanwhile, $\Delta G_3$ of Co SAs-WSe₂ appears to be significantly reduced under the negative electric fields, which rationalizes the boosted OER performance of Co SAs-WSe₂ under negative $E_{field}$. The PDOS of O 2p_z and Co 3d_{z²} orbitals (Supplementary Fig. 102) also confirms a weaker Co-O interaction under negative OEEFs (consistent with the results of $\Delta G_O$). More importantly, lower charge density (Fig. 4j) around the adsorbed *O site (Supplementary Tables 21 and 23) was observed in Co SAs-WSe₂ under negative OEEFs, which presumably promotes the consequent approaching of a OH⁻ to *O and thus reduces $\Delta G_3$ for the formation of a O-O bond, as

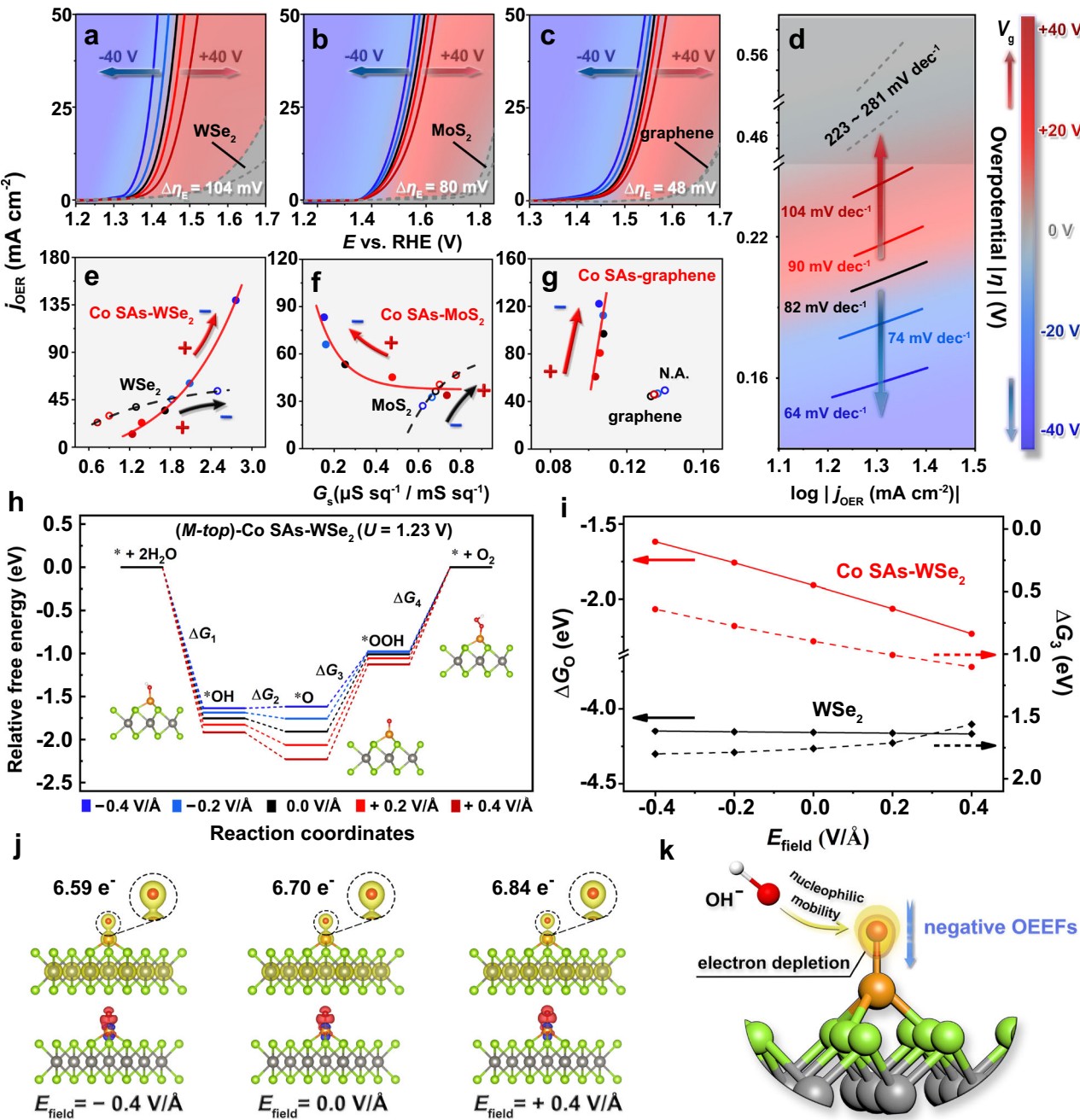

**Fig. 4 OEEF-induced "onsite electrostatic polarization" in the Co SAs-WSe₂ catalyzed OER. a–c** OER polarization curves of Co SAs-WSe₂ (**a**), Co SAs-MoS₂ (**b**), and Co SAs-graphene (**c**) in response to OEEFs. **d** Tafel curves of Co SAs-WSe₂ in response to OEEFs. The regions between two gray dashed lines in **a–d** represent the tunable OER activity of pristine 2D substrates under OEEF regulation. The color bar in **a–d** shows the direction and scope of $V_g$. **e–g** Correlations between electrochemical current density ($j_{OER}$) and in situ sheet conductance ($G_s$) of Co SAs-WSe₂ (**e**), Co SAs-MoS₂ (**f**), Co SAs-graphene (**g**), and pristine 2D substrates under OEEF regulation. The arrows in **e–g** indicate the OEEF-modulation directions of performance optimization. Note that in **e–g** the unit of abscissa is μS sq⁻¹ for Co SAs-WSe₂ and Co SAs-MoS₂, while mS sq⁻¹ for Co SAs-graphene, and the values of $j_{OER}$ in **e–g** were obtained at the potentials (vs. RHE) of: 1.45 V for Co SAs-WSe₂, 1.8 V for WSe₂, 1.6 V for Co SAs-MoS₂, 1.9 V for MoS₂, 1.6 V for Co SAs-graphene, 1.8 V for graphene. **h** Adsorption free energies diagram of (*M-top*)-Co SAs-WSe₂ for OER under OEEF regulation, where *OH, *O, and *OOH are successively absorbed on Co atom. **i** Comparisons of $\Delta G_O$ (solid lines) and $\Delta G_3$ (dashed lines) in 2H-WSe₂ (black) and (*M-top*)-Co SAs-WSe₂ (red), with $E_{field}$ from −0.4 V/Å to +0.4 V/Å. **j** Charge analysis of (*M-top*)-Co SAs-WSe₂ under OEEF regulation, where *O is adsorbed on Co atom. Top: charge densities with interface values of 0.15 e⁻/Bohr³. The zoomed-in area shows the charge distribution at the adsorbed *O site. Bottom: deformation charge densities with interface values of 0.01 e⁻/Bohr³. Red and blue areas denote electron accumulation and depletion, respectively. $E_{field}$ from left to right is −0.4 V/Å, 0, and +0.4 V/Å. **k** Schematic diagram of the "onsite electrostatic polarization" mechanism on Co SAs-WSe₂ for OER. OH⁻ in electrolyte tends to move towards the adsorbed *O site, where electron depletion was triggered by negative OEEFs.

shown in Fig. 4k. These results verified the presence of "onsite electrostatic polarization" mechanism in SA-catalyzed OER. Notably, the solvation model of (M-top)-Co SAs-WSe$_2$ (Supplementary Fig. 100) exhibited more positive free energies (Supplementary Table 19) of *OH, *O, *OOH after introducing OEEFs, which is reasonable as the stabilization effect from hydrogen bonding[72,73] is weakened by OEEFs (regardless of direction). In addition, the potential-determining step ($\Delta G_3$, from *O to *OOH) remained unchanged and the trend of OEEF-modulated charge density on the adsorbed O* ($Q_O$, Supplementary Table 22) appeared to be identical to that of the gas-phase model, demonstrating that the solvent effects did not affect the conclusion of OEEF-induced "onsite electrostatic polarization" mechanism proposed here. It is also worth mentioning that the asymmetrical oxygen species (*OH and *OOH) exhibit reorientation under the external electric field (Supplementary Figs. 104–107), which is projected on the irregular (non-linear) results of free energies (Supplementary Tables 19 and 20) in response to OEEFs. In contrast to a single sigma bond (Co-O bond in OER and Pt-H bond in HER) that is parallel to the electric field, the alteration of orientation angle leads to changes in spin states or valence states of Co SAs, contributes to the more sophisticated modulation of orbital interaction in oxygen electrocatalysis[74–76] from an alternative perspective. Overall, we have proposed and verified the "onsite electrostatic polarization" mechanism in single-atom electrocatalysis (in both cathodic and anodic reactions) that principally mimics the function of electrostatic polarization in natural enzymes, thus achieving the superior electrocatalytic performance.

**Superior electrocatalytic performance of OEEF-modulated 2D SACs.** To verify the general OEEF modulation in single-atom electrocatalysis, a series of SACs with different SA species and 2D substrates were systematically tested in the micro-platform for HER and OER, as summarized in Fig. 5a, e. Particularly, Pt SAs-MoS$_2$ ($V_g = +40$ V) exhibits excellent HER performance in acidic condition (0.5 M H$_2$SO$_4$) with the overpotential of 20 mV at 10 mA cm$^{-2}$ (Fig. 5i and Supplementary Table 9) and an ultra-high mass activity of 1171.1 A mg$_{Pt}^{-1}$ ($\eta_{HER} = -0.05$ V), which is superior to several representative Pt-based electrocatalysts in recent works (see Fig. 5k and Supplementary Tables 5 and 6)[32,33,37,47,57,59,77–81]. Moreover, due to the relatively low atomic loading in Co SACs (quantitatively derived from STEM characterizations), Co SAs-WSe$_2$ ($V_g = -40$ V) demonstrates a remarkable OER performance in alkaline condition (0.1 M KOH), with the overpotential of 139 mV at 10 mA cm$^{-2}$ (lowest among other electrocatalysts in previous studies[82–92], Fig. 5j and Supplementary Table 10) and record-high mass activity of 19,024.2 A mg$_{Co}^{-1}$ ($\eta_{OER} = 0.2$ V, Fig. 5k and Supplementary Tables 7 and 8). In addition, to verify the correspondence between on-chip electrocatalytic measurements and the actual chemical production by 2D SACs under OEEF regulation, quantitative product analysis was conducted with a custom-designed on-chip electrochemical reactor (Supplementary Fig. 26). HER and OER electrolysis were performed on SACs supported by centimeter-size graphene (Pt SAs-graphene for HER and Co SAs-graphene for OER), with application of varying $V_g$. In a 4-h potentiostatic electrolysis, the gas products (H$_2$ and O$_2$) were collected and quantified by a gas chromatograph (Supplementary Figs. 82–85). As shown in Fig. 5d, h, the chemical production rates clearly demonstrated the significant modulation by the external electric fields, further confirming the OEEF-modulated electrocatalytic process observed in the micro-platform. Therefore, the superior performance via "onsite electrostatic polarization" mechanism exhibited in OEEF-modulated

SACs offers an effective and universal strategy to tailor the active sites, thus achieving the enzyme-mimic function and high efficiency in SAC-based systems and corresponding on-chip reactors.

Long-term cyclic voltammetry (CV) scans were further performed on the model SACs under the constant and optimal electric field. During a 100-cycle test, no obvious current decay was observed in Pt SAs-MoS$_2$ ($V_g = +40$ V) for HER (Fig. 5b) and Co SAs-WSe$_2$ ($V_g = -40$ V) for OER (Fig. 5f), manifesting the high stability of SACs constructed in this work, which is also consistent with the spectroscopic results and HAADF-STEM observations (Supplementary Figs. 4 and 15). Although an irreversible oxidation peak was observed in pristine WSe$_2$ at around 1.4 V (vs. RHE) in the first few scans (Supplementary Fig. 81), which should be attributed to the inherent electrochemistry of TMDs[93–95], the introduction of Co SAs effectively shifted the OER onset potential to ~1.3 V, and led to significantly enhanced stability. PL and XPS results (Supplementary Figs. 19–22) showed that there was no significant change in the structure and chemical state of Co SAs-WSe$_2$, despite a possible trace level conversion of WSe$_2$ to WO$_3$. The chemical mapping from energy-dispersive X-ray spectroscopy (STEM-EDS, Supplementary Fig. 16) further verified the predominant of selenides localized on the surface of Co SAs-WSe$_2$ after OER stability test, validating the theoretical models employed in DFT calculations. Moreover, we tested the durability of altering external electric fields acting on Pt SAs-MoS$_2$ and Co SAs-WSe$_2$, by running linear sweep voltammetry (LSV) scans with dynamically switched $V_g$. Moreover, accompanied by $V_g$ switching among 0 V, +40 V, and −40 V every ten cycles, the electrochemical current responds rapidly to the sudden change of $V_g$ and keeps relatively steady in ten consecutive cycles (Fig. 5c, g), suggesting that the facile and quantitative strategy for introducing OEEFs, is non-destructive to electrocatalysts.

In summary, we have demonstrated the experimental evidence that the electrocatalytic performance of 2D SACs can be continuously and quantitatively modulated to a record-high level via an externally applied oriented electric field. Combined with theoretical investigations, such OEEF-induced performance enhancement is revealed to distinctively originate from "onsite electrostatic polarization" in SAC systems, where electric fields have significant influences on the charge redistribution (especially at the active sites) of SACs, leading to the optimization of chemical bonding between SA and key adsorbates, the spatial distribution of outward charges, and the reaction kinetics of the rate determine steps. This precise and easy-to-operate modulation mode reveals a universal mechanism in single-atom electrocatalysis, which applies to a broader range of electrochemical reactions. The combination of homogeneously distributed SACs with atomically precise geometric/electronic structure and the dynamic modulation through OEEFs, also presents a viable and promising functional model toward the development of artificial enzyme-mimicking systems, which holds great potential for future applications.

## Methods
**Preparation of Pt SAs-MoS$_2$.** 2D MoS$_2$ nanosheets mechanically exfoliated from bulk MoS$_2$ (n-type, SPI Supplies) were transferred to the silicon wafer using magic tape (Scotch). The silicon wafer was immersed in 30 mL deionized water and ethylene glycol (≥99%, Sigma-Aldrich) mixed solution (9:1 volume ratio) followed by 1.46 mg K$_2$PtCl$_6$ (>99.95%, Admas-beta) was added into the solution with stirring, the concentration of K$_2$PtCl$_6$ was 0.1 mmol L$^{-1}$. Then the silicon wafer immersed in the mixed solution was irradiated under the 365 nm ultraviolet light at room temperature for 2 h, which was taken out to heat at 150 °C for 15 min. Finally, Pt SAs-MoS$_2$ samples were obtained on the silicon wafer, washed by deionized water three times, and dried by N$_2$ purging. Principally, UV-light radiation induces the formation of EG radicals to reduce metal ions into single atoms[96], which appeared to be more efficient to attach the single atoms on the

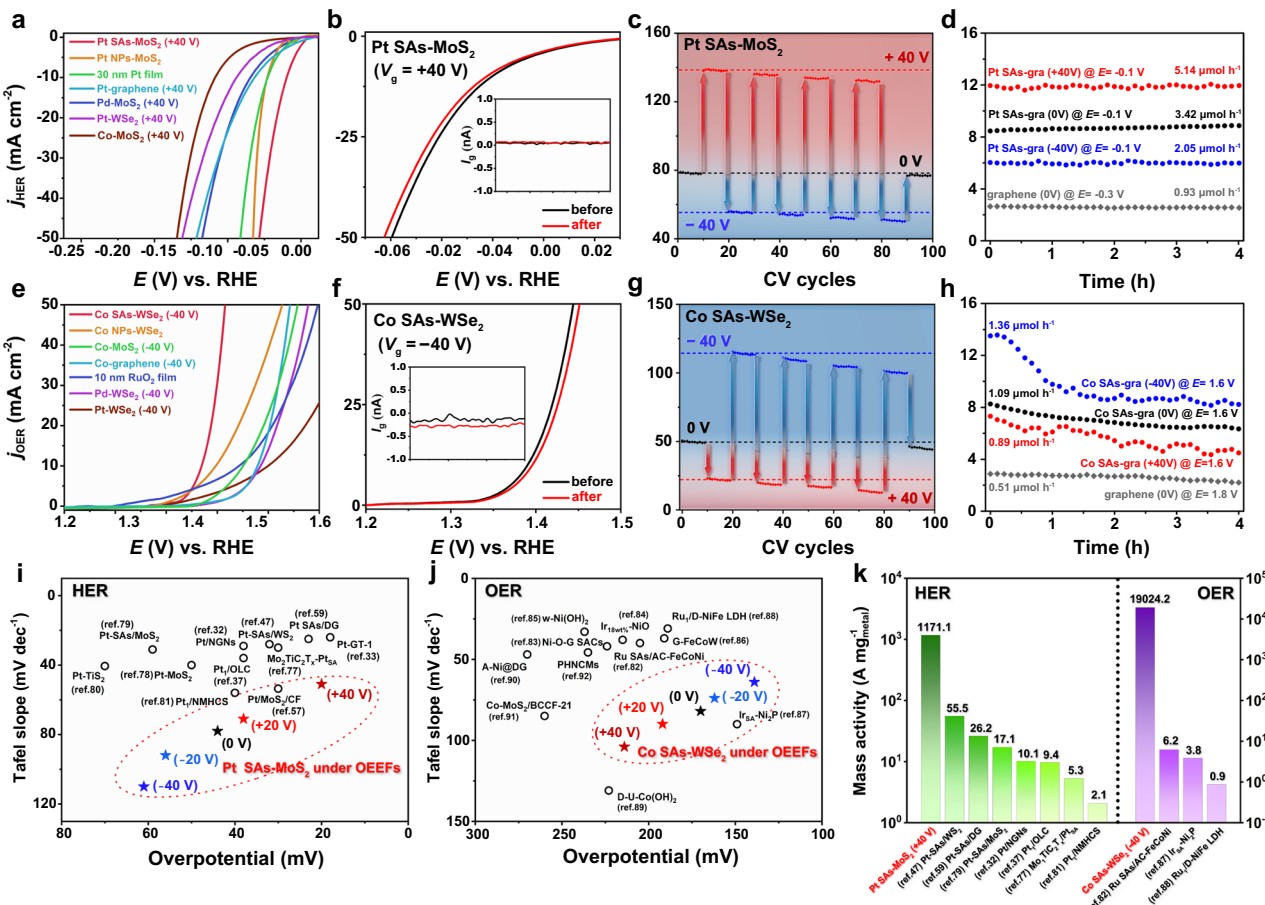

**Fig. 5 Electrocatalytic performance and stability of model SACs through "onsite electrostatic polarization" modulation. a** HER polarization curves of 2D SACs (at optimal OEEFs) and the control catalysts. **b** HER polarization curves of Pt SAs-MoS$_2$ ($V_g = +40$ V) before (black) and after (red) 100 CV cycles, in the potential range of $-0.1$ V to 0.1 V (vs. RHE). Inset (with identical abscissa) depicts the recorded leakage current. **c** Electrochemical current density ($j_{HER}$) of Pt SAs-MoS$_2$ at $E = -0.1$ V (vs. RHE) recorded in a 100-cycle HER test, with $V_g$ switching between $-40$ V and $+40$ V every ten cycles. **d** Chronoamperometric curves in a 4-h potentiostatic HER test of pristine graphene and Pt SAs-graphene under OEEF regulation. The formation rates of H$_2$ are added alone each curve. Note that the ordinates in **a–d** all correspond to HER current density ($j_{HER}$). **e** OER polarization curves of 2D SACs (at optimal OEEFs) and the control catalysts. **f** OER polarization curves of Co SAs-WSe$_2$ ($V_g = -40$ V) before (black) and after (red) 100 CV cycles, in the potential range of 1.1 V to 1.5 V (vs. RHE). Inset (with identical abscissa) depicts the recorded leakage current. **g** Electrochemical current density ($j_{OER}$) of Co SAs-WSe$_2$ at $E = 1.45$ V (vs. RHE) recorded in a 100-cycle OER test, with $V_g$ switching between $-40$ V and $+40$ V every ten cycles. **h** Chronoamperometric curves in a 4-h potentiostatic OER test of pristine graphene and Co SAs-graphene under OEEF regulation. The formation rates of O$_2$ are added alone each curve. Note that the ordinates in **e–h** all correspond to OER current density ($j_{OER}$). **i** Summarized HER performance of Pt SAs-MoS$_2$ (with varying $V_g$) and previously reported electrocatalysts. **j** Summarized OER performance of Co SAs-WSe$_2$ (with varying $V_g$) and previously reported electrocatalysts. **k** Mass activity of 2D SACs (at optimal OEEFs) in this work and several representative up-to-date catalysts. The green bars for HER at $\eta_{HER} = -0.05$ V and the purple bars for OER at $\eta_{OER} = 0.2$ V.

exfoliated pristine 2D samples (with few defect sites), see more details in Supplementary Figs. 5 and 6.

**Preparation of Co SAs-WSe$_2$.** 2D WSe$_2$ nanosheets mechanically exfoliated from bulk WSe$_2$ (p-type, 2D Semiconductors) were transferred to the silicon wafer using magic tape. The silicon wafer was immersed in 30 mL deionized water and ethylene glycol (9:1 volume ratio) mixed solution, followed by 3.57 mg CoCl$_2$·6H$_2$O (AR, Aladdin) was added into the solution with stirring, the concentration of CoCl$_2$ was 0.5 mmol L$^{-1}$. Then the silicon wafer immersed in the mixed solution was irradiated under the 365 nm ultraviolet light at room temperature for 3 h, which was taken out to heat at 150 °C for 15 min. Finally, Co SAs-WSe$_2$ samples were obtained on the silicon wafer, washed by deionized water three times, and dried by N$_2$ purging.

**Fabrication of micro-device (microcell) for electronic/electrochemical measurement.** A silicon wafer with 275 nm SiO$_2$ was cleaned and spun with a layer of photoresist (AZ5214, Nalgene). The as-prepared silicon wafer was patterned by ultraviolet lithography and deposited with Ti (20 nm)/Au (50 nm) as metal electrodes through physical vapor deposition (PVD), followed by the transfer of 2D materials and the wet impregnation for single-atom loadings. Then the silicon

wafer was spun with a layer of polymethylmethacrylate (495 PMMA A8, Microchem), patterned by e-beam lithography (eBL), and deposited with Ti (20 nm)/Au (50 nm) as the source and drain electrodes, which directly contact the 2D SACs. After lifting off the layer of PMMA, we obtained the individual micro-device for electrical measurement. Based on the fabrication of the individual micro-device, a new PMMA film was spun on the silicon wafer as the passivation layer, where a window was patterned and developed to expose the fixed area (20–100 μm$^2$) of 2D supports. The electrolyte was dripped in the window to directly contact the electrocatalysts. A graphite rod and a leak-free Ag/AgCl electrode (with 3.4 M KCl) inserted into the droplet were used as the counter electrode (CE) and the reference electrode (RE), respectively. And the 2D SACs connected with the source and drain electrodes were used as the working electrode (WE). Meanwhile, a self-reliant Au electrode (contacting the silicon directly), was fabricated as the gate electrode. To this end, the four-electrode microcell was set up for electronic/electrochemical measurement. See more details in Supplementary Figs. 23–25.

**On-chip in situ electronic/electrochemical measurement under OEEF regulation.** A three-channel system consisting of three source measurement units (SMU) from two device analyzers (Agilent B2902A) was designed for the in situ measurements. SMU1 and SMU2 from one analyzer were used for collecting the

electronic and electrochemical signals during the electrocatalytic process, and SMU3 from the other one was used for the control of gate voltages ($V_g$). In detail, SMU1 is regarded as an electrochemical workstation to control the potential of WE as to RE, and collect the electrochemical current ($I_e$) through CE. SMU2 channel is used for adding a small bias ($V_{ds}$) between the source and drain electrodes to collect the electronic signals ($I_{ds}$) through WE. SMU3 is used for applying variable $V_g$ (through the gate electrode) to introduce OEEFs on 2D materials. Notably, $I_e$ is generally several orders of magnitude smaller than $I_{ds}$, indicating that the electronic signals cannot be affected by the electrochemical current. And the tiny leakage current ($I_g < 1$ nA) flowing gate electrode can be also neglected. In typical cyclic voltammetry (CV) and linear sweep voltammetry (LSV) measurements, 0.5 M $H_2SO_4$ was used for HER and 0.1 M KOH was used for OER, with the scan rate of 20 mV s$^{-1}$. $I_e$ was recorded by SMU1 and the electrochemical current density ($j$) was calculated through normalizing $I_e$ with the exposed area of 2D materials. $I_{ds}$ was recorded by SMU2 and the sheet conductance ($G_s$) is given by,

$$G_s = \frac{1}{R_s} = \frac{L}{W} \times \frac{I_{ds}}{V_{ds}} \qquad (1)$$

where $R_s$ is the sheet resistance, $L$ is the channel length, and $W$ is the channel width.

We express the electrochemical reference voltage with respect to RHE,

$$E_{RHE} = E_{Ag/AgCl} + 0.2046V + 0.059 \text{ pH} \qquad (2)$$

Notably, area density and mass density of 2D SACs were obtained through the statistical analysis of STEM observations, to determine their mass activities (more details in Supplementary Tables 1 and 4). Besides, to accurately evaluate the electrocatalytic performance of 2D SACs proposed in this work, electrochemical measurements for the determination of catalytic activities (in Fig. 5) were conducted without operating the in situ electrical transport measurement, to avoid the potential deviations of work electrodes (micro-devices) induced by small bias ($V_{ds}$). The bias-dependent electrochemical potential deviations had been shown in Supplementary Figs. 77 and 78.

**On-chip electrochemical reactor with OEEF modulation and gas product analysis**. The CVD-grown large-size graphene (centimeter-level) was further employed to support single-atom catalysts and was fabricated into on-chip micro-device reactor, with a gas-tight PDMS placed on top as electrolyte cell and gas collection chamber. The electrolyte was injected into the cavity of PDMS to directly contact the exposed area (10 mm$^2$) of 2D SACs. A graphite rod and a leak-free Ag/AgCl electrode (with 3.4 M KCl) were used as the counter electrode (CE) and the reference electrode (RE), respectively. The SACs (on 2D graphene) connected to the Au electrodes were regarded as the working electrode (WE), and a self-reliant Au electrode was also fabricated as the gate electrode, which contacts the silicon directly. In this three-electrode system, a series of electrochemical tests were performed through a CHI630E electrochemical workstation, with gas products collected by a syringe after electrolysis. In the potentiostatic HER tests, the applied potential (vs. RHE) is: −0.3 V for graphene, −0.1 V for Pt SAs-graphene. In the potentiostatic OER tests, the applied potential (vs. RHE) is: 1.8 V for graphene, 1.6 V for Co SAs-graphene. The detection of $H_2$ and $O_2$ collected in the electrochemical reactor was analyzed by a Shimadzu GC-2030 gas chromatograph. See more details in Supplementary Figs. 26 and 82–85.

**Material characterizations**. The atomic resolution STEM images and STEM-EDS elemental mappings were recorded on an aberration-corrected STEM Titan cubed with a field emission gun at 300 kV. The beam current was 38 pA and the probe convergence semi-angle was 20.5 mrad and the angular range of the HAADF detector was from 60.5 to 200 mrad. SEM images were obtained on Hitachi S-4800 with samples deposited on carbon conductive tapes. The thickness characterizations of micro-devices were obtained on Bruker Dimension Icon atomic force microscope (AFM). The X-ray absorption data at the Pt $L$-edge of samples were recorded in the fluorescent mode at beamline 14W1[97] of the Shanghai Synchrotron Radiation Facility (SSRF), China. The station was operated with a Si (111) double-crystal monochromator at the energy of 3.5 GeV and the photon energy was calibrated with the first inflection point in Pt $L$-edge of Pt foil. X-ray photoelectron spectroscopy (XPS) was measured on a UlVAC-PHI 5000 Versa Probe spectrometer with Al-Kα as the radiation source. The binding energies in recorded spectra were calibrated by the C 1s peak, the internal standard reference at 284.6 eV. Raman and photoluminescence (PL) spectroscopies were performed on HORIBA XploRA Plus Raman microscope with a 532 nm laser. XRD patterns were recorded on a Shimadzu Lab X/XRD-6000 X-ray diffractometer equipped with a Cu-Kα radiation source ($\lambda = 0.15418$ nm) operating at 40 kV and 30 mA. The electrical measurements were performed on Lakeshore TTPX probe station. The detection of gas products collected in the electrochemical reactor was realized by a Shimadzu GC-2030 gas chromatograph.

**Theoretical calculations**. All the DFT calculations in this work were performed by using the Vienna ab initio simulation package (VASP)[98,99]. The projected augmented wave (PAW) method[100] with an energy cutoff of 500 eV was selected to describe the ion-electron interactions. The electron exchange-correlation

interactions were represented by generalized gradient approximation (GGA) with the Perdew–Burke–Ernzerhof (PBE) functional[101]. Dipole correction[102] and spin polarization were also considered during all calculations. The $k$-point mesh for structural optimization was sampled using a $3 \times 3 \times 1$ Monkhorst–Pack grid[103]. The structural optimization was terminated as the absolute force on each atom was smaller than 0.02 eV/Å and the electronic minimization reached the convergence criterion of $1 \times 10^{-6}$ eV. Bader charge calculations were used to analyze the charge transfer[104]. $\Delta G_H$ is commonly used to indicate the HER performance. A catalyst with a "$|\Delta G_H| \rightarrow 0$" might be a good HER catalyst and the hydrogen adsorption free energy ($\Delta G_H$) is computed using the equation $\Delta G_H = \Delta E_H + \Delta E_{ZPE} + \Delta U^{0 \rightarrow T} - T\Delta S$[65]. $\Delta E_H$ is the hydrogen adsorption energy computed by the equation $\Delta E_H = E(^*H) - E(^*) - 0.5E(H_2)$, where $E(^*H)$ and $E(^*)$ are the total energies of the model with and without hydrogen adsorbed on the binding site $^*$, respectively, and $E(H_2)$ is the total energy of the single hydrogen molecule. $\Delta E_{ZPE}$ is the zero-point energy difference and $T$ is the temperature (298.15 K in this study). $\Delta U^{0 \rightarrow T}$ is the internal energy at temperature $T$ and $\Delta S$ means the entropy change caused by the adsorbates. Detailed calculation equations and values of ZPE, $TS$, and $\Delta U^{0 \rightarrow T}$ are given in Supplementary Tables 12 and 17. The Gibbs free energy profiles including three oxygen intermediates (*OH, *O, and *OOH)[105] were also calculated for OER systems using a similar method. The overall OER process ($2H_2O \rightarrow O_2 + 4H^+ + 4e^-$) can be divided into four electron reaction paths under standard conditions (pH = 0, U = 0 V) shown as follows,

$$H_2O(l) + {}^* \rightarrow {}^*OH + H^+(aq) + e^-$$

$$^*OH \rightarrow {}^*O + H^+(aq) + e^-$$

$$H_2O(l) + {}^*O \rightarrow {}^*OOH + H^+(aq) + e^-$$

$$^*OOH \rightarrow {}^* + O_2(g) + H^+(aq) + e^-$$

where * represents the adsorption site, l represents the liquid phase, aq represents the solution, and g represents the gas phase, respectively. The Gibbs free energies of intermediates were denoted as $\Delta G_{OH}$, $\Delta G_O$, and $\Delta G_{OOH}$, respectively. Then, the free energy differences of each step could be calculated by equations as $\Delta G_1 = \Delta G_{OH}$, $\Delta G_2 = \Delta G_O - \Delta G_{OH}$, $\Delta G_3 = \Delta G_{OOH} - \Delta G_O$, and $\Delta G_4 = 4.92 - \Delta G_{OOH}$. The overpotential $\eta$ of OER process at $U = 0$ V can be calculated using the following equation,

$$\eta_{OER} = \max\{\Delta G_1, \Delta G_2, \Delta G_3, \Delta G_4\}/e - 1.23V \qquad (3)$$

The potential-determining step is assigned to the step corresponding to the maximum of free energy difference. Dehydrogenation of *OOH is thermodynamically easy to occur in our research systems, so $\Delta G_4$ was not considered.

The explicit solvation model with adsorbed water molecules[106] was constructed to investigate the OEEF-induced effect from solvents at the interface. For HER, one water molecule was put on top of the adsorbed *H in Pt SAs-MoS$_2$. Furthermore, two water molecules were placed around the oxygen intermediates (*OH, *O, and *OOH) in Co SAs-WSe$_2$ for OER to consider the influence of short-ranged intermolecular hydrogen bonding interactions, with more details shown in Supplementary Information.

## Data availability

The authors declare that the data supporting the conclusions of this study are available within the paper and its Supplementary Materials. Additional data are available from the corresponding authors upon reasonable request. Source data are provided with this paper.

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

## Acknowledgements

Y.P. and M.D. acknowledge the support of the National Natural Science Foundation of China (22172075 and 92156024), the Fundamental Research Funds for the Central Universities in China (020514380195 and 14380273), and the Beijing National Laboratory for Molecular Sciences (BNLMS202107). X.W. and J.M. were supported by the National Key Research and Development Program of China (2019YFC0408303) and the National Natural Science Foundation of China (21873045 and 22033004). The authors thank beamline BL14W1 (Shanghai Synchrotron Radiation Facility, SSRF) for providing the beam time.

## Author contributions

M.D. and Y.P. designed the research. X.D., J.M., P.W., L.G., and M.D. supervised the research. Y.P. designed the experiments, synthesized SACs, conducted the in situ electronic/electrochemical measurements, and performed related characterizations (spectroscopic characterizations, electrical measurements, AFM and GC analysis). Z.M., B.T., and Y.S. participated in micro-device preparation. Y.P. and M.F. transferred the 2D SACs to the copper mesh for STEM characterizations. W.Z. performed the STEM characterizations and EDS elemental mappings. W.Z., Y.P., and P.W. were responsible for STEM analysis and single-atom statistics. L.G. and H.S. provided CVD-grown monolayer MoS₂ and large-size graphene. C.L. synthesized the Pt NPs-MoS₂ and Co NPs-WSe₂ samples and performed SEM characterizations. Z.M. and Y.P. fabricated Pt film and RuO₂ film as microelectrodes and designed the on-chip electrochemical reactor for gas collection. X.W. and J.M. designed and performed the theoretical modeling work and DFT calculations. Y.P., X.W., and L.T. performed data analysis and drew the pictures. Y.P., X.W., M.D., J.M., and X.D. drafted the paper. All authors discussed the results and commented on the manuscript.

## Competing interests

The authors declare no competing interests.
