## [Peer Review File · Nature Communications]

Boosting the performance of single-atom catalysts via external electric-field polarizationREVIEWER COMMENTS

Reviewer #1 (Remarks to the Author):

The manuscript provides a detailed analysis of the role of electric fields on single atom catalyst (SAC) for HER and OER chemistry. The manuscript includes multiple methods of characterization of structural distribution of metal atoms, indicating single atom active sites. Additionally, the electronic behavior of the catalytic layer was adequately and thoroughly characterized. While the results are supported and promising, the manuscript is missing a major set of characterization which is chemical. While HER and OER has been published in many journals without measuring the rate of H₂ and O₂ formed in these reactions, it is critical that the rate of chemical formation be measured when looking at new electrocatalytic systems. In this case, the rate of H₂ and O₂ can be measured with a gas chromatograph or mass spectrometer, but it this is required for the effective study of a chemical reaction and should not be published without this basic information

Reviewer #2 (Remarks to the Author):

In this work, the authors reported an external electric field accelerated HER/OER for single-atom catalysts. Although this seems an interesting paper, many questions and concerns obviously exist (shown below). Considering the lack of technical characterizations for the structure of single-atom catalysts, the failure in reveal of real active in OER, potentially mistaken DFT models, and lacking in novelty, I cannot recommend it for publication in Nature Communications. Some detailed comments are listed below:

1. The whole work did not show well-characterized structure of single-atom catalysts. The XAS and XPS data for single-atom metals are not provided. What is the valence/chemical state and coordination environment (coordinating atom and numbers) of single-atom metals? This is compulsory and indispensable for single-atom characterizations, and essential for the further construction of DFT model.
2. TMDs materials tend to be irreversibly oxidized at around 0.7 V (vs RHE) (10.1021/acs.chemrev.5b00287). Under the onset potential of more than 1.3 V in OER, the catalysts in principle are irreversibly oxidized and no longer themselves. That also means in this case, the DFT model using WSe₂ as the substrate in OER is totally wrong. The authors should confirm the real active sites for OER and give the first few scans in OER.
3. The authors should provide the HAADF-STEM images and XPS characterizations after stability tests, especially for OER. The XPS characterization after OER measurements can be used to determine the oxidation of TMDs materials (ACS Nano 2020, 14, 4141–4152, Fig. S12), along with valence state of single-atom metals.
4. Throughout this manuscript, I think the author tried to oversell the single-atom nanozyme. The single-atom catalysts are not novel, and have been reported before (see below relevant references). Combined with the widely-used back gate voltages (Adv. Mater. 2017, 29, 1604464; Nano Lett. 2017, 17, 4109; Nano Lett. 2019, 19, 9, 6118–6123), the author found the enhancement of electrocatalysis. The extra energy in electrocatalysis, termed as overpotential, above the required thermodynamic potential is commonly applied to force the energy level of electrocatalysts comparable with the redox potential of specific reactions (such as HER). This is usually realized just through the potentiostat device, which is widely used in three-electrode-system electrocatalysis. The essence of these two methods, no matter the potentiostat or the back gate, is similar. Therefore, the concept of this work requires a more reasonable and detailed explanation. The findings connecting to the nanozyme should be well discussed.
5. Many relevant and critical references are missing. For example: Nature Communications volume 10, Article number: 5231 (2019); Nature Nanotechnology volume 13, pages411–417 (2018); Nature Communications volume 12, Article number: 3021 (2021); Energy Environ. Sci., 2015,8, 1594-1601; 10.1126/sciadv.aav5490.
6. Minor question: 1) In fig. S12, the data for S 2p XPS is missing; 2) Is UV necessary for the

fabrication of single-atom metals? For example, in Nature Nanotechnology volume 13, pages 411–417 (2018), the author just used MoS₂ to directly reduce Pt from K₂PtCl₆ because they stated that Pt atoms replaced Mo atoms in MoS₂ nanosheets spontaneously without any reductant.

Reviewer #3 (Remarks to the Author):

This manuscript presents a detailed study on the electronic nature of single atom catalysts (SACs) for HER and OER reactions and concludes by proposing that an “onsite electrostatic polarization” of the SAC occurs and leads to enhanced activity. This study combines a suite of experimental techniques and DFT modeling, including free energy, density of states, and Bader charge analyses, to determine the underlying mechanism. Overall, I find this study to be of significant interest and is suitable for publication in Nature Communications once my concerns have been addressed (see below).

1. The DFT models appear to be performed in the absence of any H₂O solvent molecules, and no mention is made as to how the OEEF would influence both the solvent molecules’ orientation at the interface and thus the system free energies. The authors need to comment on this lack in their modeling work and estimate the size of the free energy effects from such solvent induced changes. This will be particularly important to the OER discussion, as all intermediates (i.e. O*, OH*, and OOH*) will participate in hydrogen bonding with solvent at the interface.
2. The methods by which the free energies for H*, O*, OH*, and OOH* are calculated must be provided in greater detail. For example, the authors note in their methods section how the free energy of H* is calculated, but do not specifically note how the entropy term for the H* adsorption is calculated. Furthermore, nowhere do the authors state how the free energies of O*, OH*, or OOH* are calculated. This is critical information.
3. The PDOS presented in Figures S81 and S88 are discussed within the main text and used to rationalize the change in H* and O* adsorption free energy with OEEF. In particular, on page 9 the authors state “More H 1s-Pt 5dz₂ antibonding states are pulled up above Fermi level when applying positive E_{field}”. However, this effect is not visible in the PDOS figures themselves as all graphs are identical. To clarify this point and firmly establish these electronic effects, the authors need to better quantify the changes in the PDOS, **which can be accomplished via numerical integration or another computational analysis.**

Point-by-Point response to Reviewers

Reviewer #1:

Comment 1: *The manuscript provides a detailed analysis of the role of electric fields on single atom catalyst (SAC) for HER and OER chemistry. The manuscript includes multiple methods of characterization of structural distribution of metal atoms, indicating single atom active sites. Additionally, the electronic behavior of the catalytic layer was adequately and thoroughly characterized. While the results are supported and promising, the manuscript is missing a major set of characterization which is chemical. While HER and OER has been published in many journals without measuring the rate of H₂ and O₂ formed in these reactions, it is critical that the rate of chemical formation be measured when looking at new electrocatalytic systems. In this case, the rate of H₂ and O₂ can be measured with a gas chromatograph or mass spectrometer; but it this is required for the effective study of a chemical reaction and should not be published without this basic information.*

Response: We greatly appreciate the reviewer's positive comments on our study, and thank for the insightful suggestions that have inspired us to conduct key measurements on the subject. Per reviewer's suggestions, we have added additional product analysis data along with more detailed and thorough discussions, which we believe have further strengthened the foundation of our theory and improved the quality of this paper.

For a new electrocatalytic system especially performed in a micro-platform, it is indeed critical to give precise measurement of production rate of the chemicals during the electrocatalytic process, which is also important to verify the electrocatalytic regulation mechanism we propose here. This type of characterization remains rare in the electrocatalytic micro-device studies, and the on-chip production analysis will further benefit the corresponding research field. Inspired by the reviewer's suggestion, we designed an on-chip electrochemical micro-reactor enabling the collection of gas products, which could be quantified by a gas chromatograph. **To ensure that the concentration of gas products (H₂ and O₂) can reach the detection limit of the gas chromatograph, we scaled up the on-chip devices by loading SACs on a large-size graphene up to centimeter-level**, and set up an electrochemical reactor with a gas-tight PDMS as the cell for electrolyte holding and gas collection, as illustrated in **new Fig. S26**. We then performed a series of electrochemical tests (using a typical electrochemical workstation CHI630E) on the SACs (Pt SAs-graphene for HER and Co SAs-graphene for OER) with a fixed exposure area of 10 mm², and a syringe was used to collect the gas products from the upper cavity of PDMS after electrolysis (up to 4 hours) for gas chromatograph analysis. Meanwhile, OEEFs can be applied to these devices to modulate the electrocatalytic process by the same means as in other micro-device tests.

Similar to our initial experimental results on typical SACs micro-devices, OEEF regulation of the

electrocatalytic performance was clearly observed in the SACs anchored on large-size graphene, as shown in LSV curves (new Fig. S82a, S83a). The corresponding current densities matched well to those in micro-device tests, with a slight decrease which was probably due to the higher inner resistance of the large graphene sheet. More importantly, gas products collected in a four-hour potentiostatic electrolysis were successfully detected and quantified through a gas chromatograph (Shimadzu GC-2030). The formation rates of H₂ produced on Pt SAs-graphene and the formation rates of O₂ produced by Co SAs-graphene can be therefore determined, and the results are shown in chronoamperometric curves in new Figure 5d, 5h (also shown in new Figure S82b, S83b), and the original GC analysis results are shown in new Fig. S84, S85. Specifically, Pt SAs-graphene in HER exhibited the highest H₂ formation rate of 5.14 μmol h⁻¹ at -0.1 V (vs. RHE) under a positive OEEF ($V_g = +40$ V), as compared to negative OEEF ($V_g = -40$ V, 2.05 μmol h⁻¹) and no electric-field regulation ($V_g = 0$ V, 3.42 μmol h⁻¹). As control, pristine graphene showed trace H₂ production that cannot be detected at -0.1 V (vs. RHE), and a detectable H₂ production rate of 0.93 μmol h⁻¹ can only be observed at -0.3 V (vs. RHE). Similarly, for Co SAs-graphene catalyzed OER, the O₂ production rate reached up to 1.36 μmol h⁻¹ under negative OEEF regulation ($V_g = -40$ V), which is obviously higher compared to positive and no OEEF application ($V_g = +40$ V, 0.89 μmol h⁻¹; $V_g = 0$ V, 1.09 μmol h⁻¹), and graphene control ($V_g = 0$ V, 0.51 μmol h⁻¹). It is worth mentioning that in this set-up, the leakage current (I_g) monitored in the potentiostatic electrolysis was below 1 nA (new Fig. S82c, S83c), which is negligible compared with the faradaic current in the electrochemical channel, indicating no systematic/instrumental interference between the two channels. These results are all consistent with the trend in OEEF-regulated performance initially observed in the micro-device platform, replenishing a major set of chemical characterization that further support the “onsite electrostatic polarization” mechanism. To sum up, we added these additional product quantification and analysis results in new Figure 5d, 5h, Fig. S26, S82-S85 in the revised manuscript and SI, accompanied with the corresponding discussions, as the following:

On page 17, line 12, “In addition, to verify the correspondence between on-chip electrocatalytic measurements and the actual chemical production by 2D SACs under OEEF regulation, quantitative product analysis was conducted with a custom-designed on-chip electrochemical reactor (Fig. S26). HER and OER electrolysis were performed on SACs supported by centimeter-size graphene (Pt SAs-graphene for HER and Co SAs-graphene for OER), with application of varying V_g . In a 4-hour potentiostatic electrolysis, the gas products (H₂ and O₂) were collected and quantified by a gas chromatograph (Figs. S82-85). As shown in Figs. 5d and 5h, the chemical production rates clearly demonstrated the significant modulation by the external electric fields, further confirming the OEEF-modulated electrocatalytic process observed in micro-platform. Therefore, the superior performance *via* “onsite electrostatic polarization” mechanism exhibited in OEEF-modulated SACs offers an effective and universal strategy to tailor the

active sites, thus achieving the enzyme-mimic function and high-efficiency in SAC-based systems and corresponding on-chip reactors.”

On page 23, Methods section: “On-chip electrochemical reactor with OEEF modulation and gas product analysis. The CVD-grown large-size graphene (centimeter-level) were further employed to support single-atom catalysts and Pt were fabricated into on-chip micro-device reactor, with a gas-tight PDMS placed on top as electrolyte cell and gas collection chamber. Electrolyte was injected into the cavity of PDMS to directly contact the exposed area (10 mm²) of 2D SACs. A graphite rod and a leak-free Ag/AgCl electrode (with 3.4 M KCl) were used as the counter electrode (CE) and the reference electrode (RE), respectively. The SACs (on 2D graphene) connected to the Au electrodes was regarded as the working electrode (WE), and a self-reliant Au electrode was also fabricated as the gate electrode, which contacts the silicon directly. In this three-electrode system, a series of electrochemical tests were performed through a CHI630E electrochemical workstation, with the gas products collected by a syringe after electrolysis. The detection of gas products collected in the electrochemical reactor was analyzed by a Shimadzu GC-2030 gas chromatograph. See more details in Figs. S26, S82-85.”

Figure 5. Electrochemical performance and stability of model SACs under “onsite electrostatic polarization” modulation. **a**, HER polarization curves of 2D SACs (at optimal OEEFs) and the control catalysts. **b**, HER polarization curves (in 0.5 M H₂SO₄) of Pt SAs-MoS₂ ($V_g = +40$ V) before (black) and after (red) 100 CV cycles, in the potential range of -0.1 V to 0.1 V (vs. RHE) at the scan rate of $20 \text{ mV} \cdot \text{s}^{-1}$. Inset (with identical X-axis) depicts the correspondingly recorded leakage current through back-gate electrode. **c**, Electrochemical current density (j_{HER}) of Pt SAs-MoS₂ @ $E = -0.1$ V (vs. RHE) recorded in a 100-cycle HER test, with V_g switching between -40 V and

+40 V every ten cycles. **d**, chronoamperometric curves in a 4-h potentiostatic HER test of centimeter-size graphene and Pt SAs-graphene samples under OEEF regulation. The formation rates of H₂ product determined by gas chromatography are added along each curve. **e**, OER polarization curves of 2D SACs (at optimal OEEFs) and the control catalysts. **f**, OER polarization curves (in 0.1 M KOH) of Co SAs-WSe₂ ($V_g = -40$ V) before (black) and after (red) 100 CV cycles, in the range of 1.1 V to 1.5 V (vs. RHE) at the scan rate of 20 mV·s⁻¹. Inset (with identical X-axis) depicts the correspondingly recorded leakage current through back-gate electrode. **g**, Electrochemical current density (j_{OER}) of Co SAs-WSe₂ @ $E = 1.45$ V (vs. RHE) recorded in a 100-cycle OER test, with V_g switching between -40 V and +40 V every ten cycles. **h**, chronoamperometric curves in a 4-h potentiostatic OER test of centimeter-size graphene and Co SAs-graphene samples under OEEF regulation. The formation rates of O₂ product determined by gas chromatography are added along each curve. **i**, Summarized HER performance of Pt SAs-MoS₂ (with varying V_g) and previously reported electrocatalysts. **j**, Summarized OER performance of Co SAs-WSe₂ ((with varying V_g) and previously reported electrocatalysts. **k**, Mass activity of 2D SACs (at optimal OEEFs) in this work and several representative up-to-date catalysts. The green bars for HER at $\eta_{\text{HER}} = -0.05$ V and the purple bars for OER at $\eta_{\text{OER}} = 0.2$ V (also see **Tables S5-S8**).

New Figure S26. The schematic diagram (a) and optical images (b,c) of the electrochemical reactor for gas collection.

New Figure S82. HER polarization curves (a), chronoamperometric curves (b) and corresponding back-gate leakage current measurements (c) in a 4-h potentiostatic electrolysis test (right) of centimeter-size graphene and Pt SAs-graphene samples under OEEF regulation. The formation rates of H₂ product determined by gas chromatography analysis are added along each curve in **b**.

New Figure S83. OER polarization curves (a), chronoamperometric curves (b) and corresponding back-gate leakage current measurements (c) in a 4-h potentiostatic electrolysis test (right) of centimeter-size graphene and Co SAs-graphene samples under OEEF regulation. The formation rates of O₂ product determined by gas chromatography analysis are added along each curve in b.

New Figure S84. GC analysis results of the gas products from a 4-h potentiostatic electrolysis of graphene and Pt SAs-graphene for HER (shown in Fig. S82). The H₂ peak has a retention time of around 0.5 min.

New Figure S85. Corresponding GC analysis results of gas products from a 4-h potentiostatic electrolysis of graphene and Co SAs-graphene for OER (shown in Fig. S83). The O₂ peak has a retention time of around 0.9 min.

Reviewer #2:

In this work, the authors reported an external electric field accelerated HER/OER for single-atom catalysts. Although this seems an interesting paper, many questions and concerns obviously exist (shown below). Considering the lack of technical characterizations for the structure of single-atom catalysts, the failure in reveal of real active in OER, potentially mistaken DFT models, and lacking in novelty, I cannot recommend it for publication in Nature Communications. Some detailed comments are listed below:

Response: We appreciate reviewer's interest in this work and thank the reviewer for the valuable comments. We agree with the reviewer's points that more general structural and chemical characterizations, although difficult for our specific experimental platform that has not been adapted in previous investigations on single-atom catalysts, are critical to derive accurate model for DFT calculation and mechanistic conclusions. In order to address these issues, additional XPS, EXAFS, DFT calculations and product analysis were conducted, which indeed provided solid evidence to the major conclusion in this manuscript (as shown in detail below). Moreover, in light of reviewer's concerns, discussions regarding the key insights, novelty and significance of this work were updated to give better clarification, which we hope will become appealing to the reviewer. Below is the detailed point-to-point response to reviewer's comments.

Comment 1: *The whole work did not show well-characterized structure of single-atom catalysts. The XAS and XPS data for single-atom metals are not provided. What is the valence/chemical state and coordination environment (coordinating atom and numbers) of single-atom metals? This is compulsory and indispensable for single-atom characterizations, and essential for the further construction of DFT model.*

Response: We thank the reviewer for pointing out this important issue. We agree with the reviewer's point that the XAS and XPS are the most frequently employed, convincing and necessary characterizations in the studies of SACs that can reveal the structural and chemical information at the single-atom sites. To this end, SACs samples loaded on bulk 2D TMDs materials (natural crystals that was used for mechanical exfoliation micro-device fabrication) was prepared *via* the same wet-impregnation method, and the XPS and XAS characterizations were conducted. **Indeed, the combination of XPS, XANES, EXAFS and STEM allowed us to acquire the precise information related to the valance state and coordination environment of single-atom metals**, and the corresponding results have been updated in the revised manuscript and supplementary information as **new Fig. 1f-1j, Fig. S11, S20 and Table S3, S11**, and the relevant detailed discussions are as follows.

XPS characterizations were first performed to probe the electronic structure of Pt species in Pt SAs-MoS₂, as

well as control samples of Pt nanoparticles supported on MoS₂ (see Fig. S50) and commercial Pt/C. As shown in new Fig. 1h, the binding energies of Pt 4f_{7/2} in Pt SAs-MoS₂ showed a significant increase to approximately 72.5-73.0 eV, which is consistent with single atoms anchored on MoS₂ (*Nat. Commun.* 2021, 12, 3021), as compared to that of commercial Pt/C and Pt NPs-MoS₂ located at ~71.0 eV corresponding to Pt⁰. Moreover, quantitative peak deconvolution further demonstrated that the Pt species in Pt SAs-MoS₂ were in a partially oxidized valence state (Pt^{δ+}) mainly composed of Pt²⁺ and Pt⁴⁺ states, with binding energies of Pt 4f_{7/2} at 72.3 eV and 73.4 eV, respectively (see new Fig. S11). This characteristic can be attributed to the electronic interactions between Pt SAs and MoS₂ substrates (*Nat. Commun.* 2021, 12, 3021). The binding energies of Pt 4f_{7/2} of Pt SAs-MoS₂ still keeps at the same level (~72.5 eV) after HER stability test, indicating a stable coordination environment of Pt SAs anchored on MoS₂ during the electrocatalytic process.

The chemical configuration and local coordination environment of Pt SAs-MoS₂ were further investigated through X-ray absorption near-edge spectroscopy (XANES) and extended X-ray absorption fine structure (EXAFS), using the Shanghai Synchrotron Radiation Facility (SSRF) operated with a Si(111) double-crystal monochromator at energy of 3.5 GeV in the fluorescent mode (*Nucl. Sci. Tech.* 2015, 26, 050102). The Pt L₃-edge XANES profiles (new Fig. 1i) showed that the white line intensity for Pt SAs-MoS₂ was higher than that for Pt foil, indicating that the Pt species in Pt SAs-MoS₂ were oxidized. As shown by EXAFS in R space (new Fig. 1j), Pt SAs-MoS₂ exhibited a peak at approximately 2.3 Å from the Pt-S shell corresponding to a coordination number of 4.0 (new Table S3). In addition, no typical peaks for Pt-Pt bond at longer distances (> 2.7 Å) was detected, revealing the atomic dispersion of Pt atoms on MoS₂. More importantly, the EXAFS results agreed well with the theoretical distance of Pt-S bond from the optimized (*M-top*)-Pt SAs-MoS₂ model (2.29 Å, see new Table S11), and the combination of XPS, XANES, EXAFS and STEM characterizations confirms the accuracy of the model of Pt SAs anchored on *Mo-top* site of MoS₂ used in the following DFT calculations and discussions.

Similarly, XPS and XAS characterizations were also conducted on Co SAs-WSe₂ samples. Because of the relatively lower mass loading of Co SAs on WSe₂ (~0.78 wt% as determined by HAADF-STEM imaging, see Table S4), the Co K-edge XANES from Co SAs-WSe₂ samples could not be obtained, and relatively weak XPS signals of Co 2p (new Fig. S20) were acquired. As compared to that of Co NPs-WSe₂ sample (located at 777.8 eV), the Co 2p_{3/2} peak of Co SAs-WSe₂ (located at 779.3 eV) was closer to Co²⁺ reference (located at 780.3 eV, acquired using CoCl₂ sample), indicating that the Co species in Co SAs-WSe₂ exhibited a partial oxidation state (Co^{δ+}) apparently different from Co nanoparticles on WSe₂. This characteristic is consistent with Co single atom samples. Although Co K-edge XANES could not be obtained to give comprehensive information on the coordination environment at the Co single

atom sites, the combination of XPS and STEM data were sufficient to identify the existence of single atom Co and partially reveal its electronic structure, especially with previous confirmation on Pt-based SAC samples prepared via the same approach. We would also like to argue at this point that **this paper is not about a typical study on the synthesis or development of SAC materials on bulk scale. To the best of our knowledge, SACs anchored on 2D supports have not previously been integrated into a micro-device platform before this work.** One benefit of the micro-device investigation is that it enables the precise characterization of the same piece of micro-scale materials (SACs-2D) that has been actually tested, and combination of single-atom sites and mechanically exfoliated 2D atomic crystals allowed for near-perfect crystal lattice with atomic precision. **In such case, HAADF-STEM results (e.g., see original Fig. S13 and new Fig. S15) clearly showed the single-atom dispersion in our samples, where all Co SAs overlapped with W or Se atoms.** Therefore, *X-sub* or *M-top* structure should be the two reasonable models for Co SAs-WSe₂ for subsequent theoretical analysis (also note that *X-sub* and *M-top* were the thermodynamically favorable models as revealed by DFT calculations). Therefore, we had detailed discussions on each of the two optimal models of Co SAs-WSe₂, and acquired similar theoretical results (see new Figs. S101, S103 and new Table S19, Original Table S20, S21, S23, updated also in response to the suggestions from Reviewer #3) to verify the existence of “onsite electrostatic polarization” mechanism in the OER process, regardless of the exact bonding models between Co SA and the WSe₂ support.

With all above-mentioned additional new results (new Fig. 1f-1j, Fig. S11, S20 and Table S3, S11), the corresponding discussions were also added into the revised manuscript and SI, as following:

In the revised manuscript, page 3, line 20: “Aberration-corrected high-angle annular dark-field STEM (HAADF-STEM), which has been widely adopted in microstructure characterization³⁹, demonstrated that Pt SAs were successfully introduced into the 2H-MoS₂ lattice without disturbing its crystalline structure (Fig. 1b-1e and Figs. S1-S4). The Mo:S atomic ratios (Table S2) are determined to be 1.00:1.98 and 1.00:1.96 in bulk MoS₂ and Pt SAs-MoS₂, respectively, further confirming that the near-perfect lattice (few defects) was maintained. The area density of Pt SAs on mechanically exfoliated MoS₂ is determined to be 0.42 atom·nm⁻² via statistical analysis of the STEM images (Table S1). Raman (Fig. 1f) and photoluminescence spectroscopy (Fig. S8) revealed the doping effect of Pt SAs on 2H-MoS₂ nanosheets⁴⁰⁻⁴³, which also leads to a more positive threshold voltage in the electrical FET transfer curves (Fig. 1g and Fig. S7) of n-type MoS₂ micro-device modified by Pt SAs⁴⁴⁻⁴⁶.”

The chemical configuration and local coordination environment of Pt SAs-MoS₂ were then investigated through the combination of X-ray photoelectron spectroscopy (XPS) and X-ray absorption spectroscopy (XAS). Compared to that of commercial Pt/C and Pt NPs-MoS₂ (Fig. S50 and S51) located at around 71.0 eV (for Pt⁰), the Pt 4f_{7/2} XPS peak (Fig. 1h) of Pt SAs-MoS₂ significantly moved towards higher binding energies (approximately 72.5-73.0 eV),

with obvious alteration in peak shape. Moreover, the results of peak deconvolution further demonstrated that the Pt species in Pt SAs-MoS₂ were partially oxidized to the valence state of 0-+4 (Pt^{δ+}), mainly composed of Pt²⁺ and Pt⁴⁺ (72.3 eV and 73.4 eV, **Fig. S11**), which should be attributed to the electronic interactions between Pt SAs and MoS₂ nanosheets⁴⁷. Moreover, in X-ray absorption near-edge spectroscopy (XANES) the Pt L₃-edge analysis (**Fig. 1i**) showed that the white line intensity for Pt SAs-MoS₂ was higher than that for Pt foil, indicating the oxidized Pt species were not in the metallic state⁴⁷⁻⁴⁹. As shown by EXAFS in R space (**Fig. 1j**), Pt SAs-MoS₂ exhibited a peak at approximately 2.3 Å from the Pt-S shell with a coordination number of 4.0 (**Table S3**). Besides, no typical peaks for Pt-Pt bond at longer distances (>2.7 Å) was detected, revealing the atomic dispersion of Pt atoms on MoS₂ nanosheets. More importantly, the theoretical distance of Pt-S bond (2.291 Å, **Table S11**) from the optimized model of (*M-top*)-Pt SAs-MoS₂ (more details in **Fig. S88**) was also consistent with the EXAFS results, further confirming the Mo-atop site of Pt SAs anchored on MoS₂.”

On page 13, line 4: “Co SAs-WSe₂ was used as model catalyst as it combines an OER favorable SAC and a p-type substrate. As confirmed by HAADF-STEM images (**Figs. S12-14**), Co SAs were successfully introduced into the mechanically exfoliated WSe₂, with the area density of 0.41 atom·nm⁻² (~0.78 wt%, see **Table S4**). The Co 2p XPS signals (**Fig. S20**) also indicated that the oxidized Co species (Co^{δ+}) in Co SAs-WSe₂ is different from Co nanoparticles.”

Line 18: “All possible configuration of Co atom anchored on 2H-WSe₂ were considered and three DFT models were constructed (see **Fig. S98**). In line with the STEM observations, the two optimal models with lower formation energies (*M-top* and *X-sub*) were ultimately applied for subsequent theoretical analysis through introducing different E_{field} , with successive *OH, *O and *OOH adsorptions. **Figure 4h** shows the adsorption free energies (ΔG_{OH} , ΔG_{O} , ΔG_{OOH}) of three oxygen intermediates along the overall OER pathway on (*M-top*)-Co SAs-WSe₂. Specifically, the adsorption free energies can be effectively tailored by the external electric field, accompanied with obvious changes in free energy differences (ΔG_1 , ΔG_2 , ΔG_3 , ΔG_4). And the maximum of free energy difference (ΔG_3 , see **Methods** and **Tables S19, S20**) indicated that the conversion from *O to *OOH was the potential-determining step (pds) in Co SAs-WSe₂ for OER. As shown in **Figure 4i**, ΔG_{O} of (*M-top*)-Co SAs-WSe₂ move towards the zero point when E_{field} goes negatively (red solid line), whereas WSe₂ has no obvious change in ΔG_{O} (black solid line). Meanwhile, ΔG_3 of Co SAs-WSe₂ appears to be significantly reduced under the negative electric fields, which rationalizes the boosted OER performance of Co SAs-WSe₂ under negative E_{field} .”

New Figure 1. Synthesis and characterizations of Pt SAs-MoS₂. **a**, Schematic illustration for the preparation of Pt SAs-MoS₂ micro-device. **b,c**, HAADF-STEM images of mechanically exfoliated monolayer MoS₂. **c** is the magnified image of yellow dashed area in **b**. **d,e**, HAADF-STEM images of Pt SAs-MoS₂. **e** is the magnified image of yellow dashed area in **d**. Red circles highlight the Pt SAs distributed in the 2H-MoS₂ lattice. **f**, Raman spectroscopy of mechanically exfoliated monolayer MoS₂ (black) and Pt SAs-MoS₂ (red). **g**, Transfer characteristics of FET devices fabricated from mechanically exfoliated monolayer MoS₂ (black) and Pt SAs-MoS₂ (red). **h**, Peak deconvolution of Pt 4f XPS spectra for commercial Pt/C, Pt NPs-MoS₂, Pt SAs-MoS₂. **i**, Normalized Pt L₃-edge XANES spectra of Pt foil, Pt SAs-MoS₂ and PtO₂. **j**, First shell fitting of EXAFS spectra for Pt foil, Pt SAs-MoS₂ and PtO₂.

New Figure S11. XPS spectra showing Pt 4f core level peak regions for K_2PtCl_4 (Pt^{2+}) and K_2PtCl_6 (Pt^{4+}) samples.

New Table S3. The first shell fitting parameters of the EXAFS results for Pt SAs-MoS₂ and Pt foil.

Sample	Scatter	CN	R/Å	$\sigma^2/\text{Å}^2$	$\Delta E_0/\text{eV}$
Pt SAs-MoS ₂	Pt-S	4.0	2.292	0.0073	4.964
Pt foil	Pt-Pt	12	2.765	0.0042	8.207
PtO ₂	Pt-O	6.7	1.916	0.0028	10.872

R, distance between absorber and backscatter atoms. CN, coordination number. σ^2 , Debye-Waller factor. ΔE_0 , inner potential correction to account for the difference in the inner potential between the sample and the reference compound.

New Table S11. Bond length of Pt atom binds to MoS₂ in different DFT models.

Model	X-sub	M-top	h-top
Binding form	Pt-Mo	Pt-S	Pt-S
Bond length	2.632	2.291	2.343

New Figure S20. XPS spectra showing Co 2p core level peak regions for CoCl₂ (black), Co NPs-WSe₂ (blue), Co SAs-WSe₂ before (red) and after OER stability test (magenta).

Comment 2: TMDs materials tend to be irreversibly oxidized at around 0.7 V (vs RHE) (10.1021/acs.chemrev.5b00287). Under the onset potential of more than 1.3 V in OER, the catalysts in principle are irreversibly oxidized and no longer themselves. That also means in this case, the DFT model using WSe₂ as the substrate in OER is totally wrong. The authors should confirm the real active sites for OER and give the first few scans in OER.

Response: We gratefully thank the reviewer for raising this important issue regarding the stability of TMDs materials during the OER process, which indeed determines the accuracy of the DFT and electrostatic polarization model proposed in the following studies. Per reviewer's suggestion, additional electrochemical stability tests (with long-term cyclic voltammetry scans) were performed on 2D TMDs before and after the introduction of single-atom metals (Pt SAs-MoS₂ for HER and Co SAs-WSe₂ OER) with the assistance of multiple electrochemical, spectroscopic and imaging characterizations were conducted to address this issue.

First we would like to point out that the HER/OER performance stability of SACs in this study were initially revealed in a 100-cycle LSV tests, as were shown in original Figure 5b and 5e. After 100 cycles of LSV tests, Pt-SAs-MoS₂ and Co SAs-WSe₂ (best catalyst in each reaction) showed only slight decline in the HER and OER activities, respectively, indicating the good stability of the materials during long-term electrochemical tests. Per reviewer's suggestion, the first few scans in a 100-cycle LSV test were added as **new Fig. S79** in the revised SI,

indicating that the SAC-2D material/devices remain intact within the first few LSV scans.

We also agree with the reviewer that pristine TMD supports could be irreversibly oxidized, which has been commonly accepted in a number of literatures (*Chem. Rev.* 2015, 115, 11941-11966; *Faraday Discuss.* 2008, 140, 219-231; *Surface Science* 1982, 119, 46-60). This additional data is indeed important as it indicates the robustness of all the electrochemical and transport measurement data that eventually lead to the proposed mechanistic conclusions. To give a thorough consideration on the stability issue, we further compared the initial electrochemical scans on the pristine TMD supports before and after the introduction of single atoms (as shown in **new Fig. S80, S81**), specifically on the WSe₂ that is a favorable choice in OER. It can be observed that WSe₂ demonstrated an oxidation peak at 1.4 V (vs. RHE) in our on-chip set-up, as shown in **new Fig. S81**. This peak can be attributed to the oxidation of W^{IV} in WSe₂ to W^{VI} into WO₃ (*Small* 2021, 17, 2100510; *ACS Nano* 2014, 8, 12, 12185-12198), and gradually disappeared within first several scans (left panel in **new Fig. S81**), indicating the irreversible oxidation process. Notably, the onset potential for OER moved to ~1.3 V (vs. RHE) when Co SAs were anchored onto the WSe₂ (right panel in **new Fig. S81**), which is a clear sign that the OER was initiated on the Co SA sites rather than the WSe₂. Therefore, in the typical LSV testing range of Co SA-WSe₂ (<1.45 V vs. RHE where OER current is already significant) in our study, the oxidation of WSe₂ support can be maintained at a low level, leading to negligible impact on our initial mechanistic conclusions and DFT models.

One might also argue that the low-level irreversible oxidation may occur at an even lower potential (~0.7 V vs. RHE) that was not reflected on LSV peaks, and given the slight decline in the OER performance of Co SAs-WSe₂ after 100 cycles of LSV scans, the irreversible oxidation of WSe₂ could still occur over long-time anodic testing, and indeed the accurate quantification of such low-level oxidation is challenging. Inspired by reviewer's suggestions in **Comment 3**, additional spectroscopic (XPS and PL) and STEM characterizations were further conducted on Pt SAs-MoS₂ and Co SAs-WSe₂ after HER/OER stability tests, to give a comprehensive understanding on the oxidation level in our electrocatalytic systems. The corresponding results are demonstrated and discussed more in the response to **Comment 3**.

New Figure S79. The first three scans and the last scan of Pt SAs-MoS₂ for HER (left, $V_g = +40$ V) and Co SAs-WSe₂ for OER (right, $V_g = -40$ V) in a 100-cycle LSV test.

New Figure S80. The first few scans of the pristine MoS₂ (left) and Pt SAs-MoS₂ (right) for HER.

New Figure S81. The first few scans of the pristine WSe₂ (left) and Co SAs-WSe₂ (right) for OER.

Comment 3: *The authors should provide the HAADF-STEM images and XPS characterizations after stability tests, especially for OER. The XPS characterization after OER measurements can be used to determine the oxidation of TMDs materials (ACS Nano 2020, 14, 4141–4152, Fig. S12), along with valence state of single-atom metals.*

Response: We thank the reviewer again for this valuable suggestion that allowed us to better determine the oxidation behaviors of TMD substrates and elucidate their stabilities in our electrochemical tests (including the long-term LSV stability tests). Inspired by reviewer's comments, **additional spectroscopic (XPS and PL) and STEM characterizations** were conducted on Pt SAs-MoS₂ and Co SAs-WSe₂ before and after HER/OER stability tests, to give a comprehensive understanding on the oxidation level in our electrocatalytic systems on the basis of the electrochemical results (which were demonstrated in the response to **Comment 2**).

As compared to the as-prepared sample, Pt SAs-MoS₂ showed no change in the XPS peaks of Mo 3d, S 2s, S 2p and Pt 4f (**new Fig. S10**) after HER stability test (100 cycles of LSV in the potential range of 0.1 to -0.1 V vs. RHE), indicating that there is no change in the chemical state of Pt SAs-MoS₂ during the HER process. This is also consistent with unchanged PL curves, as shown in **new Fig. S8**. The magnified HAADF-STEM images (**new Fig. S4**) after HER stability test further revealed that the 2H-MoS₂ substrate maintained the original crystal structure and phase. **These data all indicated good stability of Pt SAs-MoS₂ during the HER investigation.**

Similarly, for Co SAs-WSe₂ samples after OER stability test (100 cycles of LSV in the potential range of 1.0 to 1.45 V vs. RHE), no significant change in peak intensities was observed in the XPS peaks of W 4f, Se 3d, O 1s and Co 2p (**new Fig. S21, S22**), indicating that **the Co SAs-WSe₂ largely retains its crystal structure after the long-term OER test.** This conclusion is also consistent with the PL results, as shown in **new Fig. S19**. Close comparison revealed a very slight increase in the intensity of the W 4f peak located at around 38.0 eV (**new Fig. S22b**), which suggests the **possible formation of trace W⁶⁺ species** (*Small 2021, 17, 2100510; ACS Appl. Nano Mater. 2020, 3, 4218–4230*). As shown in **new Fig. S21**, the existence of O 1s peak indicates the original trace defect sites in the as-exfoliated pristine WSe₂ 2D lattice, which however does not contribute to the coordination environment at the single atom sites, as the same O 1s peak was observed after single atom decoration. These trace oxygens remained unchanged after long-term OER investigations, further indicating that mild oxidation in OER does not lead to the formation of new oxygen species that influence the coordination environment of single atom sites. According to the reference data analysis suggested by the reviewer (*ACS Nano 2020, 14, 4141–4152, Fig. S12*), O 1s peak was divided into O1, O2 and O3 peaks, which can be respectively assigned to the metal-O bond, lattice oxygen species and surface hydroxyl groups. As further shown in the right panel of **new Fig. S21**, the W-O bond character (indicated by O1 peak) is negligible after OER tests, which is in line with the trace W⁶⁺ 4f XPS signals. The Co 2p XPS signals (**new Fig.**

S22d) also indicated that the bonding between Co SAs and WSe₂ kept in a relatively stable form that was not affected by the OER investigations. **These above results revealed that the Co SAs-WSe₂ experienced no significant alteration in its crystal structure and chemical states during the anodic LSV tests, with only a trace level of irreversible oxidization from WSe₂ to WO₃.** This obvious difference between the oxidation level in Co SAs-WSe₂ and pristine WSe₂ samples can be attributed to the high chemical activity of the single atom sites, which largely and practically reduced the oxidation potentials in typical OER tests in this study. Importantly, **HAADF-STEM images acquired after long-term OER stability tests (see new Fig. S15) clearly revealed that the atomic structure/phase of 2H-WSe₂ in Co SAs-WSe₂ was sustained during OER.** As shown in **new Fig. S16**, STEM-EDS analysis also illustrated the predominant presence of selenides localized on the surface of Co SAs-WSe₂ after long-term OER tests, with the extremely low level of O content.

In summary, based on all above electrochemical, spectroscopic and microscopic characterization results, we can conclude that **the 2D TMD substrates used in this work maintained its original structure/phase after electrochemical (cathodic and anodic LSV) tests, thus ensuring the stable forms of structural/chemical environments for anchored single atoms and electronic interactions between metal single-atom sites and TMD substrates.** Trace level of irreversible oxidation was observed for WSe₂ during long-term OER, but the extremely low-level oxidation does not affect the above conclusions (i.e., the presumed crystal, chemical/coordination and electronic structures at the single atom sites), and **certainly does not cause any impact on the typical short-term measurements that were used to derive the “electrostatic polarization mechanism” identified in this study.** Similar stability tests and results have been also reported before, which confirmed the good stability of TMDs materials during the low intense OER process (*Adv. Mater. Interfaces* 2019, 6, 1900948; *ACS Nano* 2019, 13, 3162-3176; *Adv. Energy Mater.* 2017, 7, 1602217). For the same reason we believe the DFT models used for the *in silico* mechanistic investigations were correct, which have also been employed and discussed in literatures (*Nanoscale* 2018, 10, 20113-20119; *Adv. Mater. Interfaces* 2016, 4, 1500669).

Overall, we appreciate reviewer’s important insights and valuable suggestions, which have led us to a more systematic evaluation regarding the crystal, chemical and electronic structure of our SA-2D systems, and further verified the precision of our model, with a number of additional valuable electrochemical, spectroscopic and microscopic results. To sum up, we added these additional results as **new Figs. S4, S8, S10, S15, S16, S19-S22** in the revised manuscript and SI, accompanied with the corresponding discussions, as the following:

In revised MS, page 17, line 27: “... SACs constructed in this work, which is also consistent with the spectroscopic

results and HAADF-STEM observations (Figs. S4, S15). Although an irreversible oxidation peak was observed in pristine WSe₂ at around 1.4 V (vs. RHE) in the first few scans (Fig. S81), which should be attributed to the inherent electrochemistry of TMDs⁷⁷⁻⁷⁹, the introduction of Co SAs effectively shifted the OER onset potential to ~1.3 V (Fig. S81), and led to significantly enhanced stability. PL and XPS results (Figs. S19-22) showed that there was no significant change in structure and chemical state of Co SAs-WSe₂, despite a possible trace level conversion of WSe₂ to WO₃. The chemical mapping from energy dispersive X-ray spectroscopy (STEM-EDS, Fig. S16) further verified the predominant of selenides localized on the surface of Co SAs-WSe₂ after OER stability test, validating the theoretical models employed in DFT calculations.”

New Figure S10. XPS spectra showing Mo 3d (a), S 2p (b) and Pt 4f (c) core level peak regions for bulk MoS₂ (black) and Pt SAs-MoS₂ before (blue) and after HER stability test (red).

New Figure S8. PL spectroscopy of mechanically exfoliated MoS₂ (black) and Pt SAs-MoS₂ before (blue) and after HER stability test (red).

New Figure S4. HAADF-STEM images of Pt SAs on mechanically exfoliated MoS₂ after the HER stability test. Red circles indicated the Pt SAs.

New Figure S22. XPS spectra showing W 4f (a,b), Se 3d (c) and Co 2p (d) core level peak regions for bulk WSe₂ (black) and Co SAs-WSe₂ before (blue) and after OER stability test (red).

New Figure S21. XPS spectra (left) and peak deconvolution (right) showing O 1s core level peak regions for bulk WSe₂ and Co SAs-WSe₂ before and after OER stability test.

New Figure S19. PL spectroscopy of mechanically exfoliated WSe₂ (black) and Co SAs-WSe₂ before (blue) and after OER stability test (red).

New Figure S15. HAADF-STEM images of Co SAs on mechanically exfoliated WSe₂ after OER stability test. Red circles indicated the Co SAs.

New Figure S16. STEM-EDS elemental mappings (W, Se, Co and O) of Co SAs on mechanically exfoliated WSe₂ after OER stability test.

Comment 4: *Throughout this manuscript, I think the author tried to oversell the single-atom nanozyme. The single-atom catalysts are not novel, and have been reported before (see below relevant references). Combined with the widely-used back gate voltages (Adv. Mater. 2017, 29, 1604464; Nano Lett. 2017, 17, 4109; Nano Lett. 2019, 19, 9, 6118–6123), the author found the enhancement of electrocatalysis. The extra energy in electrocatalysis, termed as overpotential, above the required thermodynamic potential is commonly applied to force the energy level of electrocatalysts comparable with the redox potential of specific reactions (such as HER). This is usually realized just through the potentiostat device, which is widely used in three-electrode-system electrocatalysis. The essence of these two methods, no matter the potentiostat or the back gate, is similar. Therefore, the concept of this work requires a more reasonable and detailed explanation. The findings connecting to the nanozyme should be well discussed.*

Response: We thank the reviewer for pointing out the lack of clarification and possible confusion in our statements and discussions, which should be better clarified in the manuscript. First of all, we agree with the reviewer's point that both SACs and application of back-gate voltages are previously reported before. However, the use of back-gate voltage as a means to dynamically and precisely generate local OEEFs, which can effectively modulate the electrolytic activity of SACs, certainly has its own novelty and significance. To the best of our knowledge, similar experimental approaches and theoretical innovations have not been reported in the SACs systems before this study, despite the abundant works in the development of high-performance SACs. We would also like to emphasize here that **the concept of OEEF-regulated chemical reactions has emerged as an important research field, and similar**

concepts have been theoretically predicted (*J. Am. Chem. Soc.* 2018, 140, 13350-13359; *J. Am. Chem. Soc.* 2004, 126, 11746-11749), experimentally revealed (*Sci. Adv.* 2019, 5, eaaw3072; *Nature* 2016, 531, 88-91), and thoroughly discussed in reviewer and perspective publications (*Chem. Soc. Rev.* 2018, 47, 5125-5145; *Chem. Soc. Rev.* 2018, 47, 5146-5164; *ACS Catal.* 2018, 8, 5153-5174; *Nat. Chem.* 2016, 8, 1091-1098). Despite the widely-accepted importance and significance of the concept, its experimental realization has been rare, often with complicated single-molecular level instrumentations (STM-based technology, break junction single-molecule measurements, etc.). Therefore, our work here demonstrates the first successful example of OEEF-modulation in the SACs systems (and also heterogeneous electrocatalytic processes for this matter), with a relatively simple yet dynamic, quantitative and highly precise on-chip approach. In addition, the electrostatic on-site orbital polarization mechanism (that alters the catalytic pathway in HER) is also unprecedented and unique to the SACs system. This mechanism is fundamentally different from the simple Fermi level-modulation (*Adv. Mater.* 2017, 29, 1604464; *Nano Lett.* 2017, 17, 4109; *Nano Lett.* 2019, 19, 9, 6118-6123) or “self-gating” effect (*Nat. Mater.* 2019, 18, 1098-1104) in the 2D semiconductors, which are discussed with more detail below.

Second, we agree with reviewer’s point that the overpotential “is commonly applied to force the energy level of electrocatalysts comparable with the redox potential of specific reactions (such as HER)”, which is the essence in electrochemistry, and “usually realized just through the potentiostat device”. We believe that the reviewer raised a very good point that has not been thoroughly discussed in previous publications regarding the use of back-gate voltage on 2D materials (*Adv. Mater.* 2017, 29, 1604464; *Nano Lett.* 2017, 17, 4109; *Nano Lett.* 2019, 19, 9, 6118-6123). The application of the back-gate voltage essentially raises the Fermi level of semiconductor electrode towards the chemical potentials of the key intermediate states (determined by the intrinsic redox potential and overpotential of the desire redox pair conversion) to help overcome the energy barrier (E_a) and drive the electrochemical reactions. This is basically similar to the working principle of applying electrochemical potential (through potentiostat) itself in terms of energy diagram consideration, and in terms of physics, both methods lead to the injection of electrons with higher energy for the reduction of H^+ in HER. However, we would like to argue that one big difference is that **the back-gate voltage remains at a steady state and does not contribute to electron/energy transfer processes**. In such set-up, no current (electrical nor Faradic) was passing through the back-gate circuit, as the charge accumulated at the MoS_2-SiO_2 interface kept stable during the electrochemical process. Therefore, there is no energy input by the application of back-gate voltage. Meanwhile, the apparent overpotential and hence the electrolysis voltage was reduced in the electrochemical circuit (top-gate), and the overall energy input is reduced in this system. Overall, the on-chip field-effect semiconductors modulation of electrochemical reactions can be understood like this: **the back-**

gate voltage raises the Fermi level of semiconductor electrode via a circuit separate from the top-gate electrochemical circuit without inducing the back-gate current (due to the existence of SiO₂ dielectric layer and lack of MoS₂-SiO₂ interfacial charge transfer during back-gate modulated electrochemical processes), therefore this part practically reduces the overall energy input that is needed from the top-gate electrochemical channel.

It should also be noted that in addition to the change of Fermi level, which is to overcome the intrinsic *chemical overpotentials*, the application of gate voltage also has the effect of tailoring the carrier concentrations of the semiconductors and change their conductivities. This effect is especially obvious when the semiconductor is initially in an “off” state, causing a large inner resistance and therefore *physical resistance overpotential*. Without the application of back-gate voltage, the conductivity issue is also to be addressed by applying the electrochemical potentials, now known as “self-gating” effect (*Nat. Mater.* 2019, 18, 1098–1104). With the application of back-gate voltage (*Adv. Mater.* 2017, 29, 1604464; *Nano Lett.* 2019, 19, 6118–6123; *Nano Lett.* 2019, 19, 7293–7300), the semiconductor was first “turned on” to a conductive state, therefore no additional electrochemical potential (top-gate voltage) was needed to overcome this part of physical overpotential caused by large inner resistance. This phenomenon was also observed in our study, in addition to the above-mentioned results, we further demonstrated the opposite tailoring direction of p-type WSe₂ in HER, where the positive back-gate voltage theoretically raised the Fermi level to reduce the chemical overpotential, yet the apparent reaction was not promoted (but hindered) because the p-type WSe₂ was turned off, and large resistance lead to a significantly increased physical overpotential (Fig. 2e and 2h in the manuscript).

It can be summarized that the back-gate voltage serves as the means to modulate both Fermi level and the inner resistance of the 2D semiconductors. On this basis, we further demonstrate in this paper, for the first time, that the back-gate modulation can effectively utilized to tailor the activity of single-atom catalytic sites. The SACs system cannot be simply recognized as a semiconductor material, and we demonstrate that the modulation of Fermi level or inner resistance does not fully match the modulation trend on the SAC activities. Original Fig. 2 shows the inconsistent correlation between SACs with different types of 2D supports, and original Fig. 3 shows that while DFT calculated hydrogen binding energy (ΔG_{H} , which is typically recognized as the energy barrier in HER) agrees with the back-gate induced Fermi level change, yet their trend with applied back-gate voltage does not match to the modulation of HER activity. We therefore conclude that the back-gate voltage introduces a local electrostatic field at the single-atom sites, which polarizes its atomic orbitals and electron distributions. Such electrical-field induced polarization consequently alters the rate determine step in HER, and therefore pushes the catalytic process

out of the limitation of the volcano plot, as summarized in Fig. 3e. Again, as there is no charge transfer across the MoS₂-SiO₂ interface, the back-gate voltage practically introduced a steady electrostatic field during the electrochemical processes, which does not consume additional energy while lowering down the electrochemical overpotential at the top-gate channel. To better clarify this issue, the correspondingly recorded back-gate current during our electrochemical and ETS measurements were added as **new Figs. 5b, 5f** in the revised MS and **new Figs. S82, S83, S86, S87** in the revised SI.

Overall, we gratefully thank the reviewer for raising up such an inspiring point that brought new insights into the mechanistic considerations both to our work and regarding the previously reported literatures. Inspired by the reviewer, additional data and discussions were added in the revised manuscript and SI to better clarify this matter. Along with these new discussions, we hope we can also convince the reviewer that our investigation has its own novelty and significance in terms of experimental realization of the OEEF-modulation on SACs, as well as the unique modulation mechanism regarding the catalytic pathways and reaction kinetics.

On page 6, line 18, "... and gradually approaches Pt-based electrocatalysts reported previously⁵⁷⁻⁵⁹. **Note that negligible back-gate leakage current (Figs. S86, S87) was observed, indicating charge accumulated at the MoS₂-SiO₂ interface was kept stable during the top-gate electrochemical process, and no extra energy input is required by the introduction of an electrostatic field via back-gate circuit.**"

On page 6, line 30, "First, a positive and close-to-linear $j_{\text{HER}}-G_s$ correlation (black dashed line in **Fig. 2g**) was established in pristine MoS₂. **In this case, carrier (electron) injection into n-type semiconductors can lead to the gradually declined overpotential originated from inner resistance (contact and/or sheet resistance of MoS₂), and the rise of their Fermi level reduces the corresponding energy barrier in HER (*i.e.*, change of H adsorption ability), thus promoting the HER kinetics^{17, 18, 61}.**"

In addition, we also agree with the reviewer that the focused statement of "nanozymes" in the original manuscript might be an oversell on the concept. In fact, the novel "electrostatic polarization" mechanism on SACs induced by OEEFs itself should be the major point of this study. Although there have been proposals to link the structural similarity between SACs and nanozyme, and similar OEEFs modulation has been summarized in natural enzyme systems, we can leave to the readers in the correlated, multi-disciplinary research fields to decide whether or not the combination of SACs and OEEFs should be a "nanozyme" system. To tune down this unnecessary emphasis, the title of was changed to "**Boosting the performance of single-atom catalysts via external electric-field polarization**", and several statements regarding "nanozyme" concepts have been deleted in the revised paper.

New Figure S86. HER polarization curves (left) of Pt SAs-MoS₂ ($V_g=+40$ V) before and after 100 CV cycles, and the correspondingly recorded leakage current (right) flowing gate electrode.

New Figure S87. OER polarization curves (left) of Co SAs-WSe₂ ($V_g=-40$ V) before and after 100 CV cycles, and the correspondingly recorded leakage current (right) flowing gate electrode.

Comment 5: Many relevant and critical references are missing. For example: *Nature Communications* volume 10, Article number: 5231 (2019); *Nature Nanotechnology* volume 13, pages411–417 (2018); *Nature Communications* volume 12, Article number: 3021 (2021); *Energy Environ. Sci.*, 2015,8, 1594-1601; 10.1126/sciadv.aav5490.

Response: We thank the reviewer for pointing out this issue of incomplete reference citation, the corresponding references have been added in the revised manuscript and SI. Indeed, these relevant works have motivated us to conduct further experiments and theoretical simulations, which were discussed exhaustively in the response to other comments from the reviewer. Particularly, we added the **new References 26, 47, 48, 49** as the research background on nanozyme-like SACs ([26] *Sci. Adv.* 2019, 5, eaav5490), to support the up-to-date characterizations on 2D SACs in our work ([47] *Nat. Commun.* 2021, 12, 3021; [48] *Nat. Nanotech.* 2018, 13, 411; [49] *Energy Environ. Sci.* 2015, 8, 1594), and to make electrochemical performance comparison (*Nat. Commun.* 2021, 12, 3021; *Nat. Commun.* 2019, 10, 5231; *Energy Environ. Sci.* 2015, 8, 1594. See **new Table S9** in the revised SI). With more balanced and appropriate reference citations, we believe the quality of manuscript is also improved.

On page 2, line 21, “Single atom catalysts (SACs), which have arguably become one of the most active frontiers in

heterogeneous catalysis for its high atom utilization efficiency^{21, 22}, featuring highly tunable active centers with well-defined coordination geometric/electronic structures similar to typical homogeneous catalysts or nanozymes²³⁻²⁶.”

On page 4, line 3: “Moreover, the results of peak deconvolution further demonstrated that the Pt species in Pt SAs-MoS₂ were partially oxidized to the valence state of 0-+4 (Pt^{δ+}), mainly composed of Pt²⁺ and Pt⁴⁺ (72.3 eV and 73.4 eV, **Fig. S11**), which should be attributed to the electronic interactions between Pt SAs and MoS₂ nanosheets⁴⁷. Moreover, in X-ray absorption near-edge spectroscopy (XANES) the Pt L₃-edge analysis (**Fig. 1i**) showed that the white line intensity for Pt SAs-MoS₂ was higher than that for Pt foil, indicating the oxidized Pt species were not in the metallic state⁴⁷⁻⁴⁹.”

Comment 6: *Minor question: 1) In fig. S12, the data for S 2p XPS is missing; 2) Is UV necessary for the fabrication of single-atom metals? For example, in Nature Nanotechnology volume 13, pages411–417 (2018), the author just used MoS₂ to directly reduce Pt from K₂PtCl₆ because they stated that Pt atoms replaced Mo atoms in MoS₂ nanosheets spontaneously without any reductants.*

Response: 1) We thank the reviewer for pointing out the issue of missing XPS data. The XPS data for S 2p of bulk MoS₂ and Pt SAs-MoS₂ were added in the revised SI as new Fig. S10, along with the complete XPS data of bulk WSe₂ and Co SAs-WSe₂ in new Fig. S22.

2) We thank the reviewer for this question regarding the choice of synthetic method in this work. We did attempt to synthesize the Pt SACs in a wet-impregnation method without adding any reductants or UV-light. However, the resulted sample exhibited poor HER performance similar to the freshly prepared MoS₂ nanosheets, indicating the absence of Pt species (corresponding HER-LSV curves were added as new Fig. S5). Inspired by the reported synthesis of Pd SACs in *Science*, 2016, 352, 797-800, we then used ethylene glycol (EG), which could converse into EG radicals under UV radiation, as the reducing agent to acquire Pt SAs. Notably, when UV was not applied in this route (just with addition of EG), the resulting samples only had a slightly promoted HER performance. Only Pt SA-MoS₂ samples prepared under 365 nm UV radiation exhibited significantly enhanced HER performance comparable to commercial Pt/C. We further performed STEM characterizations on Pt-MoS₂ samples without reactants or without UV radiation, in which the density of Pt single atoms was obviously lower than the samples under 365 nm UV radiation, as shown in new Fig. S6. All these data verified that the UV-assisted reduction process is necessary for synthesizing 2D SACs proposed in this work, and the corresponding results were added in the revised SI to better clarify the choice of synthetic route in this work.

As for the article *Nature Nanotechnology* 2018, 13, 411–417, the author used MoS₂ to directly reduce K₂PtCl₆ into Pt SAs in a wet-impregnation method, and stated that Pt atoms replaced Mo atoms in MoS₂ nanosheets

spontaneously. These results seem to be different from what we observed in our study, which can be rationalized by the different level of defects of the 2D materials. In particular, MoS₂ used in *Nat. Nanotech.* work was synthesized through a hydrothermal method, which generally induces a relatively high level of defect sites. The defects created a mixed valence and thereby endowed MoS₂ with redox ability (*Nat. Commun.* 2018, 9, 2120; *Nat. Mater.* 2016, 15, 48-53) to better trap single atoms. However, the mechanically exfoliated 2D substrates in this study came from the natural crystals and had much lower defect level (as confirmed by HAADF-STEM images and XPS results, see Fig. S1 and Table S2) that could not effectively trap the anchor the single atoms without the addition of reducing agent. Moreover, as confirmed by the added analysis of XANES and EXAFS (new Fig. 1g, 1h), the Pt SAs were attached to the MoS₂ on the Mo-atop site, which was different from the doping configurations (*Mo-sub*) reported in the *Nat. Nanotech.* work. DFT calculations revealed that the *M-top* model of Pt SAs required lower formation energy (new Fig. S87 and Table S12) when binding to 2H-MoS₂, which is also consistent with the theoretical results reported before (*Energy Environ. Sci.* 2015, 8, 1594-1601). To this end, it is difficult for Pt atoms to replace Mo atoms spontaneously in MoS₂ crystals with near-perfect lattice, and thus the extra reductants were required for the synthesis of SACs in our work. In sum, new Figs. S5, S6, S10 and S22 were added in the revised SI to better clarify this issue. We added some brief discussions in the Methods section in revised manuscript, as following:

On page 21, line 24, Methods section: “Principally, UV-light radiation induces the formation of EG radicals to reduce metal ions into single atoms⁸⁰, which appeared to be more efficient to attach the single-atoms on the exfoliated pristine 2D samples (with few defect sites), see more details in Supplementary Fig. S5 and S6.”

New Figure S5. HER polarization curves of MoS₂ (black), commercial Pt/C (gray), Pt-MoS₂ without reductants (blue), without UV radiation (magenta) and under UV radiation (red).

New Figure S6. HAADF-STEM images of Pt-MoS₂ without reductants (**a**), without UV radiation (**b**) and under UV radiation (**c**).

Reviewer #3:

This manuscript presents a detailed study on the electronic nature of single atom catalysts (SACs) for HER and OER reactions and concludes by proposing that an “onsite electrostatic polarization” of the SAC occurs and leads to enhanced activity. This study combines a suite of experimental techniques and DFT modeling, including free energy, density of states, and Bader charge analyses, to determine the underlying mechanism. Overall, I find this study to be of significant interest and is suitable for publication in Nature Communications once my concerns have been addressed (see below).

Comment 1: *The DFT models appear to be performed in the absence of any H₂O solvent molecules, and no mention is made as to how the OEEF would influence both the solvent molecules’ orientation at the interface and thus the system free energies. The authors need to comment on this lack in their modeling work and estimate the size of the free energy effects from such solvent induced changes. This will be particularly important to the OER discussion, as all intermediates (i.e. O*, OH*, and OOH*) will participate in hydrogen bonding with solvent at the interface.*

Response: We thank for the reviewer to raise this valuable suggestion. It is indeed important to clarify the OEEF-induced influence of solvent molecules. In response, the solvation models with adsorbed water molecules (*Electrochimica Acta* 2010, 55, 7975-7981) were constructed to investigate the OEEF-induced effect from solvents at the interface. The adsorption free energy and Bader charge were analyzed in these solvation models and the related results were added as **new Tables. S13, S15, S19, S22 and new Figs. S90, S91, S100, S101** in the revised SI. The corresponding detailed discussion was listed as follows.

In the first step, one water molecule was introduced into the model of (*M-top*)-Pt SAs-MoS₂ (confirmed by XAS results) for HER. For solvation model, the values of ΔG_{H} become more positive (**new Table S13**) indicated a relatively weaker *H adsorption ability in presence of water, which could be originated from the energy requirement for the structural rearrangements (the orientation of Pt-H bond, see **new Fig. S90**) induced by the electrostatic effect on water molecules (*Nature* 2021, 600, 81–85; 10.1021/jacs.2c01094). In addition, ΔG_{H} of solvation model exhibited a negative correlation with the applied E_{field} (**new Fig. S91**), which is similar to that of gas-phase model. More importantly, the trend of OEEF-modulated charge density on adsorbed *H (Q_{H} , **new Table S15**) is identical to that of gas-phase model, indicating that despite the overall system energy level, introduction of solvent molecules does not alter the major conclusion in our study about charge distribution at the SA site.

To go further, two water molecules were placed around the oxygen intermediates (*OH, *O and *OOH) in the model of (*M-top*)-Co SAs-WSe₂ (consistent with the STEM observations) for OER. For solvation model (**new Fig.**

S100), the comparatively less adsorption feasibility (*OH, *O and *OOH) seen in the solvent phase was also assigned to the structural rearrangements (10.26434/chemrxiv-2021-4pq0b) at the SA site. The free energy values (ΔG_{OH} , ΔG_{O} and ΔG_{OOH} , see new Table S19) with no electric field (0.0 V/Å) were always more negative than those with the external electric field applied (-0.4, -0.2, +0.2, +0.4 V/Å), which could be rationalized by the fact that the stabilization effect from hydrogen bonding (*The Journal of Physical Chemistry C* 2017, 121, 11455-11463; *The Journal of Physical Chemistry C* 2015, 119, 1032-1037) is weakened (disturbed) by the introduction of OEEFs (regardless of direction). More importantly, the maximum free energy difference was still subordinate to ΔG_3 (new Fig. S101), indicating that the potential-determining step (PDS) of OER remained unchanged (from *O to *OOH). Additionally, the trend of OEEF-modulated charge density on adsorbed *O (Q_{O} , new Table S22) is also identical to that of gas-phase model, confirming that the free energy effects from solvent-induced changes did not interfere with the OEEF-induced “electrostatic polarization” mechanism proposed in this work.

On page 10, line 16: “... Solvation model of (*M-top*)-Pt SAs-MoS₂ was further constructed to investigate the possible effect from solvent molecules (water) at the electrochemical interface (Fig. S90). The values of hydrogen adsorption energy in presence of a water molecule exhibited a negative correlation with the introduced electric field gradients, which is similar to the $\Delta G_{\text{H}}-E_{\text{field}}$ relationship observed in gas-phase model (Table S13 and Fig. S91). More importantly, the trend of OEEF-modulated charge distribution on the adsorbed *H (Q_{H} , see Table S15) is also close to the gas-phase model.”

On page 14, line 1: “... mechanism in SA-catalyzed OER. Notably, the solvation model of (*M-top*)-Co SAs-WSe₂ (Fig. S100) exhibited more positive free energies (Fig. S19) of *OH, *O, *OOH after introducing OEEFs, which is reasonable as the stabilization effect from hydrogen bonding^{72, 73} is weakened by OEEFs (regardless of direction). In addition, the potential-determining step (ΔG_3 , from *O to *OOH) remained unchanged and the trend of OEEF-modulated charge density on the adsorbed O* (Q_{O} , see Table S22) appeared to be close to that of gas-phase model, demonstrating that the solvent effects did not affect the conclusion of OEEF-induced “onsite electrostatic polarization” mechanism proposed here.”

New Figure S90. The explicit solvation models of (*M-top*)-Pt SAs-MoS₂ (*H adsorbed), with E_{field} applied from -0.4 V/Å to +0.4 V/Å.

New Table S13. Hydrogen adsorption free energies (ΔG_{H} , $U=0$ V) of 2H-MoS₂ (Mo atom) and (*M-top*)-Pt SAs-MoS₂ (Pt atom, gas-phase and solvation models), with E_{field} applied from -0.4 V/Å to $+0.4$ V/Å.

ΔG_{H} (eV)		-0.4 V/Å	-0.2 V/Å	0.0 V/Å	$+0.2$ V/Å	$+0.4$ V/Å
2H-MoS ₂		0.047	0.049	0.051	0.053	0.055
(M-top)-Pt SAs-MoS ₂	gas-phase	0.098	0.040	-0.025	-0.092	-0.165
	solvation	0.373	0.360	0.250	0.171	0.078

New Figure S91. Hydrogen adsorption energy (ΔG_{H}) in the non-solvation and solvation models of (*M-top*)-Pt SAs-MoS₂.

New Table S15. Bader charge of the adsorbed *H (Q_{H}) in the gas-phase and solvation models of (*M-top*)-Pt SAs-MoS₂, with E_{field} applied from -0.4 V/Å to $+0.4$ V/Å.

Q_{H}	-0.4 V/Å	0.0 V/Å	$+0.4$ V/Å
gas-phase	1.944	2.008	2.080
solvation	1.964	1.989	2.070

New Figure S100. The explicit solvation models of (*M-top*)-Co SAs-WSe₂ (*OH, *O, and *OOH adsorbed), with E_{field} applied from -0.4 V/\AA to $+0.4 \text{ V/\AA}$.

New Table S19. Adsorption free energies of oxygen intermediates (*OH, *O, and *OOH, $U=1.23 \text{ V}$) and free energy differences (ΔG_1 , ΔG_2 and ΔG_3) in (*M-top*)-Co SAs-WSe₂ (Co atom, gas-phase and solvation models), with E_{field} applied from -0.4 V/\AA to $+0.4 \text{ V/\AA}$.

ΔG (eV)		-0.4 V/\AA	-0.2 V/\AA	0.0 V/\AA	$+0.2 \text{ V/\AA}$	$+0.4 \text{ V/\AA}$
ΔG_{OH}	gas-phase	-1.637	-1.687	-1.753	-1.828	-1.917
	solvation	-1.215	-1.443	-1.711	-1.538	-1.559
ΔG_{O}	gas-phase	-1.618	-1.758	-1.908	-2.064	-2.231
	solvation	-1.310	-1.608	-1.902	-1.828	-1.834
ΔG_{OOH}	gas-phase	-0.977	-0.983	-1.010	-1.057	-1.127
	solvation	-0.276	-0.540	-0.824	-0.660	-0.559
$\Delta G_1 = \Delta G_{\text{OH}}$	gas-phase	-1.637	-1.687	-1.753	-1.828	-1.917
	solvation	-1.215	-1.443	-1.711	-1.538	-1.559
$\Delta G_2 = \Delta G_{\text{O}} - \Delta G_{\text{OH}}$	gas-phase	0.020	-0.071	-0.155	-0.236	-0.314
	solvation	-0.095	-0.166	-0.191	-0.290	-0.275
$\Delta G_3 = \Delta G_{\text{OOH}} - \Delta G_{\text{O}}$	gas-phase	0.641	0.776	0.898	1.008	1.103
	solvation	1.034	1.068	1.078	1.168	1.275

New Figure S101. Adsorption free energies diagrams of *(M-top)*-Co SAs-WSe₂ (gas-phase and solvation models) for OER under OEEF regulation, where *OH, *O and *OOH is successively adsorbed on the exposed Co atom.

New Table S22. Bader charge of the adsorbed *O (Q_O) in the gas-phase and solvation models of *(M-top)*-Co SAs-WSe₂, with E_{field} applied from -0.4 V/\AA to $+0.4 \text{ V/\AA}$.

models	Q_O	-0.4 V/\AA	0.0 V/\AA	$+0.4 \text{ V/\AA}$
(M-top) -Co SAs-WSe ₂	gas-phase	6.594	6.699	6.836
	solvation	6.470	6.477	6.482

Comment 2: The methods by which the free energies for H^* , O^* , OH^* , and OOH^* are calculated must be provided in greater detail. For example, the authors note in their methods section how the free energy of H^* is calculated, but do not specifically note how the entropy term for the H^* adsorption is calculated. Furthermore, nowhere do the authors state how the free energies of O^* , OH^* , or OOH^* are calculated. This is critical information.

Response: We grateful thank the reviewer for reminding us the lack of computational details. To address this issue, related critical information had been added to **Methods section** in the revised manuscript and more details in the revised SI (**new Tables. S12, S17**), which was also listed as follows.

On page 24, line 4, “**Theoretical calculations.** All the DFT calculations in this work were performed by using the Vienna ab initio simulation package (VASP)^{82, 83}. The projected augmented wave (PAW) method⁸⁴ with an energy cutoff of 500 eV was selected to describe the ion-electron interactions. The electron exchange-correlation interactions were represented by generalized gradient approximation (GGA) with the Perdew-Burke-Ernzerhof (PBE) functional⁸⁵. Dipole correction⁸⁶ and spin polarization were also considered during all calculations. The k-point mesh for structural optimization was sampled using a $3 \times 3 \times 1$ Monkhorst-Pack grid⁸⁷. The structural optimization was terminated as the absolute force on each atom was smaller than 0.02 eV/\AA and the electronic minimization reached the convergence criterion of $1 \times 10^{-6} \text{ eV}$. Bader charge calculations were used to analyze the charge transfer⁸⁸. ΔG_H is

commonly used to indicate the HER performance. A catalyst with a “ $|\Delta G_H| \rightarrow 0$ ” might be a good HER catalyst and the hydrogen adsorption free energy (ΔG_H) is computed using the equation $\Delta G_H = \Delta E_H + \Delta E_{ZPE} + \Delta U^{0 \rightarrow T} - T\Delta S$.⁶⁵ ΔE_H is the hydrogen adsorption energy computed by equation $\Delta E_H = E(*H) - E(*) - 0.5E(H_2)$, where $E(*H)$ and $E(*)$ are the total energies of the model with and without hydrogen adsorbed on the binding site *, respectively, and $E(H_2)$ is the total energy of the single hydrogen molecule. ΔE_{ZPE} is the zero-point energy difference and T is the temperature (298.15 K in this study). $\Delta U^{0 \rightarrow T}$ is the internal energy at temperature T and ΔS means the entropy change caused by the adsorbates. The Gibbs free energy profiles including three oxygen intermediates (*OH, *O and *OOH)⁸⁹ were also calculated for OER systems using the similar method. The overall OER process can be described as the following equation under standard conditions ($pH = 0$, $U = 0$ V),

and this equation could be divided into four electron reaction paths as equations (1-4).

where * represents the adsorption site, l represents the liquid phase, aq represents the solution, and g represents the gas phase, respectively. The Gibbs free energies of intermediates were denoted as ΔG_{OH} , ΔG_O , and ΔG_{OOH} , respectively. Then, the free energy differences of each step could be calculated by equations $\Delta G_1 = \Delta G_{OH}$, $\Delta G_2 = \Delta G_O - \Delta G_{OH}$, $\Delta G_3 = \Delta G_{OOH} - \Delta G_O$ and $\Delta G_4 = 4.92 - \Delta G_{OOH}$. The overpotential η of OER process at $U = 0$ V can be calculated using following equation,

$$\eta^{OER} = \max\{\Delta G_1, \Delta G_2, \Delta G_3, \Delta G_4\}/e - 1.23$$

The potential-determining step (pds) is assigned to the step corresponding to the maximum of free energy difference. Dehydrogenation of *OOH is thermodynamically easy to occur in our research systems, so ΔG_4 was not considered.

The explicit solvation model with adsorbed water molecules⁹⁰ was constructed to investigate the OEEF-induced effect from solvents at the interface. For HER, one water molecule was putted on top of the adsorbed *H in Pt SAs-MoS₂. Furthermore, two water molecules were placed around the oxygen intermediates (*OH, *O and *OOH) in Co SAs-WSe₂ for OER to consider the influence of short-ranged intermolecular hydrogen bonding interactions, with more details shown in **Supplementary Information.**”

New Table S12. Thermodynamic corrections to the Gibbs free energy (eV) of hydrogen intermediates at $T=298.15$ K.^a

Models	Intermediates	ZPE – TS + $\Delta U^{0 \rightarrow T}$
2H-MoS ₂	*H	0.150
(M-top)-Pt SAs-MoS ₂	*H	0.144

^a The calculated results under zero electric field were used uniformly, because electric field has negligible influence on the thermodynamic corrections.

New Table S17. Thermodynamic corrections to the Gibbs free energy (eV) of oxygen intermediates at T=298.15 K.^a

Models	Intermediates	ZPE – TS + $\Delta U^{0 \rightarrow T}$
2H-WSe ₂	*OH	0.333
	*O	0.083
	*OOH	0.260
(M-top)-Co SAs-WSe ₂	*OH	0.296
	*O	0.020
	*OOH	0.332
(X-sub)-Co SAs-WSe ₂	*OH	0.294
	*O	0.034
	*OOH	0.383

^a The calculated results under zero electric field were used uniformly, because electric field has negligible influence on the thermodynamic corrections.

Comment 3: The PDOS presented in Figures S81 and S88 are discussed within the main text and used to rationalize the change in H* and O* adsorption free energy with OEEF. In particular, on page 9 the authors state “More H 1s-Pt 5d_{z²} antibonding states are pulled up above Fermi level when applying positive E_{field}”. However, this effect is not visible in the PDOS figures themselves as all graphs are identical. To clarify this point and firmly establish these electronic effects, the authors need to better quantify the changes in the PDOS, which can be accomplished via numerical integration or another computational analysis.

Response: We thank the reviewer for pointing out this issue. The PDOS previously shown in Figures S81 and S88 were indeed not clear enough for establishing the electronic effects we stated in the manuscript. For a more clarified demonstration of the change in *H and *O adsorption energy with OEEFs, we added the details of the PDOS and quantified the OEEF-induced PDOS changes by calculating the energy gap ($\Delta\varepsilon$) between the centers of bonding states and antibonding states, which had been shown in new Figs. S92, S102 in the revised SI.

In detail, new arrows were added to clearly indicate the bonding states and antibonding states in the PDOS shown in the left of new Figs. S92, S102, and the changes in PDOS were indicated by the energy gap ($\Delta\varepsilon$) between the centers of bonding states and antibonding states. Specifically, the band centers (*Advances in catalysis* 2000, 45, 71-129) of bonding states and antibonding states were calculated with following equations, and the energy gap ($\Delta\varepsilon = \varepsilon_b - \varepsilon_{ab}$) between the centers of bonding states (ε_b) and antibonding states (ε_{ab}) center was determined.

$$\varepsilon_b = \frac{\int_b n_b(\varepsilon)\varepsilon d\varepsilon}{\int_b n_b(\varepsilon)d\varepsilon} \quad \varepsilon_{ab} = \frac{\int_{ab} n_{ab}(\varepsilon)\varepsilon d\varepsilon}{\int_{ab} n_{ab}(\varepsilon)d\varepsilon}$$

The energy gap between bonding states and antibonding states become smaller when applying positive OEEFs, which requires less energy to form the hybridized orbitals and thus results in a more compact bond (*Nature* 1995, 376, 238-240; *PNAS* 2011, 108, 3, 937-943; *Physical Review Materials* 2019, 3, 092801). To better quantify these electronic effects, the energy gap was further correlated with the adsorption free energies at different OEEFs. As shown in the right of new Figs. S92, S102, the positive near-linear $\Delta\varepsilon$ - ΔG_H ($\Delta\varepsilon$ - ΔG_O) relationship reveal that the OEEF-modulated hydrogen (oxygen) adsorption ability is consistent with the PDOS changes, which rationalizes the electronic effects induced by OEEFs.

On page 10, line 12, "... in response to E_{field} . Accordingly, the partial density of states (PDOS) projected onto the H 1s and the Pt $5d_z^2$ orbitals show consistent response to OEEFs (Fig. S92), and the energy gap between the centers of H 1s-Pt $5d_z^2$ bonding states and antibonding states become smaller when applying positive OEEFs. In this case, less energy is required to form the hybridized orbitals⁶⁷⁻⁶⁹ and thus higher *H adsorption energy on Pt-MoS₂ (a more compact Pt-H bond) is observed."

New Figure S92. PDOS (left) of H 1s orbital (blue) and Pt $5d_z^2$ orbital (red) in (*M-top*)-Pt SAs-MoS₂ at different OEEFs, showing the energy gap ($\Delta\varepsilon$) between the centers of H 1s-Pt $5d_z^2$ bonding states and antibonding states and its correlation (right) with hydrogen adsorption free energy (ΔG_H).

New Figure S102. PDOS (left) of O 2p_z orbital (green) and Co 3d_{z²} orbital (orange) in (*M-top*)-Co SAs-WSe₂ at different OEEFs, showing the energy gap ($\Delta\varepsilon$) between the centers of O 2p_z-Co 3d_{z²} bonding states and antibonding states and its correlation (right) with oxygen adsorption free energy (ΔG_O).

REVIEWER COMMENTS

Reviewer #1 (Remarks to the Author):

The authors have adequately addressed all reviewer comments

Reviewer #2 (Remarks to the Author):

I am pleased with the authors' answer to my comments and the corresponding changes brought to the manuscript. They have addressed the concerns that I raised in the initial review. In the revised version and the response to my comments, the authors have added the combination of XPS, XANES, EXAFS and STEM characterizations to confirm the precise information related to the valence state and coordination environment of single-atom metals. They also emphasize the concept of OEEF-accelerated chemical reactions at the atomic level, and highlights the novelty. Additionally, the stability of TMDs during OER has been further carefully confirmed. Therefore, it is now ready for publication.

Reviewer #3 (Remarks to the Author):

Two of my previous concerns have been thoroughly addressed by the authors. However, I still find the theoretical methods section to be lacking, ultimately making these results non-reproducible. The authors did not sufficiently expand upon the methods used to estimate the internal energy and entropy factors. The authors use the following equation " $\Delta G_H = \Delta E_H + \Delta E_{ZPE} + \Delta U - T\Delta S$ " to compute the free energy for H* adsorption. How exactly are the ΔU and ΔS terms calculated in this study. What physics theories, mathematical formalism, etc. is used here? Without a detailed methods section that enables reproduction of the work, I cannot recommend publication of this work.

Point-by-Point response to Reviewers

Reviewer #1:

The authors have adequately addressed all reviewer comments.

Response: We greatly appreciate the reviewer's positive comments.

Reviewer #2:

I am pleased with the authors' answer to my comments and the corresponding changes brought to the manuscript. They have addressed the concerns that I raised in the initial review. In the revised version and the response to my comments, the authors have added the combination of XPS, XANES, EXAFS and STEM characterizations to confirm the precise information related to the valance state and coordination environment of single-atom metals. They also emphasize the concept of OEEF-accelerated chemical reactions at the atomic level, and highlights the novelty. Additionally, the stability of TMDs during OER has been further carefully confirmed. Therefore, it is now ready for publication.

Response: We greatly appreciate the reviewer's positive comments.

Reviewer #3:

Two of my previous concerns have been thoroughly addressed by the authors. However, I still find the theoretical methods section to be lacking, ultimately making these results non-reproducible. The authors did not sufficiently expand upon the methods used to estimate the internal energy and entropy factors. The authors use the following equation " $\Delta G_H = \Delta E_H + \Delta E_{ZPE} + \Delta U - T\Delta S$ " to compute the free energy for H adsorption. How exactly are the ΔU and ΔS terms calculated in this study. What physics theories, mathematical formalism, etc. is used here? Without a detailed methods section that enables reproduction of the work, I cannot recommend publication of this work.*

Response: We greatly appreciate the reviewer's suggestion to give computational details of the free energy calculations ($\Delta G_H = \Delta E_H + \Delta E_{ZPE} + \Delta U - T\Delta S$). Accordingly, we added discussions about the mathematical formalisms of S (entropy) and $\Delta U^{0 \rightarrow T}$ (internal energy at temperature T of intermediate), and provided more detailed values in the revised SI (see page S58 and new Tables S12, S17). In fact, the values of $ZPE - TS + \Delta U^{0 \rightarrow T}$ term were given in Table S12 and Table S17 of the original version. Now in the present version we added the values of each thermodynamic correction term in Supplementary Information. In addition, we further compared our results with some other recent references, as shown in Table R1. Our results are almost identical to those reported for the similar system (*Small* 2022, 2105129). And it should be mentioned that the system investigated in a recent

literature (*Nat. Commun.* 2022, 12, 2193), in which the Mo atom is substituted by Pt atom (called *M-sub*), is quite different from (*M-top*)-Pt SAs-MoS₂ in the present work, thus resulting in a more negative value of free energy. The correlated revisions in MS and SI were also listed as follows, with yellow highlights.

Table R1. The hydrogen binding free energy reported in different systems

Models	Theoretical level	ΔG_H / eV	Thermal Correction / eV	Reference
(M-top)-Pt SAs-MoS ₂	DFT/PBE-D3	-0.025	0.143	This work
(M-top)-Pt SAs-MoS ₂	DFT/PBE	~ -0.03	NA ^a	Jiao S, et al. Small , 2022 , 2105129.
(M-sub)-Pt SAs-MoS ₂ ^b	DFT/PBE	-1.09	NA ^a	Xu J, et al. Nat. Commun. , 2022 , 12, 2193.

^a NA, which indicates the detailed values are not available in the reference.

^b The model is quite different from the above two models of (*M-top*)-Pt SAs-MoS₂, since the Mo atom is substituted by Pt atom in (*M-sub*)-Pt SAs-MoS₂ system.

on page 24, line 20 in revised MS (Method section): “ $\Delta U^{0 \rightarrow T}$ is the internal energy at temperature T and ΔS means the entropy change caused by the adsorbates. Detailed calculation equations and values of ZPE, TS and $\Delta U^{0 \rightarrow T}$ were given in Supplementary Tables S12, S17.”

On page S58 in revised SI: “The Gibbs free energy of each intermediate is given by $G = E + E_{ZPE} - TS + \Delta U^{0 \rightarrow T}$, where E is the total energy of adsorption model, and T is equal to 298.15 K. E_{ZPE} , S and $\Delta U^{0 \rightarrow T}$ are zero-point energy, entropy, and internal energy at temperature T of intermediate, respectively. The values of S and $\Delta U^{0 \rightarrow T}$ could be calculated from equations (S1) and (S2), respectively.

$$S = R \sum_i \left\{ \frac{h\nu_i}{k_B T} \frac{e^{-\frac{h\nu_i}{k_B T}}}{1 - e^{-\frac{h\nu_i}{k_B T}}} - \ln \left[1 - e^{-\frac{h\nu_i}{k_B T}} \right] \right\} \quad (S1)$$

$$\Delta U^{0 \rightarrow T} = R \sum_i \left(\frac{h\nu_i}{k_B} \right) \left(\frac{1}{2} + \frac{e^{-\frac{h\nu_i}{k_B T}}}{1 - e^{-\frac{h\nu_i}{k_B T}}} \right) \quad (S2)$$

where R is molar gas constant, k_B is Boltzmann constant, h is Plank constant, and ν_i is frequency for the i -th vibration mode. All these thermodynamic correction terms could be calculated using VASPKIT^[4]. The frequency below 50 cm⁻¹ is set to 50 cm⁻¹. The computational data for E_{ZPE} , S and $\Delta U^{0 \rightarrow T}$ are listed in Table S12 (for hydrogen intermediates) and Table S17 (for oxygen intermediates).”

New Table S12. Detailed thermodynamic corrections to the Gibbs free energy (eV) of hydrogen intermediates at T=298.15 K.^a

Models	Intermediates	ZPE	TS	$\Delta U^{0 \rightarrow T}$
--------	---------------	-----	----	------------------------------

2H-MoS ₂	*H	0.157	0.010	0.008
(M-top)-Pt SAs-MoS ₂	*H	0.162	0.046	0.027

^a The calculated results under zero electric field were used uniformly, because electric field has negligible influence on the thermodynamic corrections.

New Table S17. Thermodynamic corrections to the Gibbs free energy (eV) of oxygen intermediates at T=298.15 K.^a

Models	Intermediates	ZPE	TS	$\Delta U^{0 \rightarrow T}$
2H-WSe ₂	*OH	0.351	0.054	0.036
	*O	0.090	0.027	0.019
	*OOH	0.351	0.183	0.092
(M-top)-Co SAs-WSe ₂	*OH	0.341	0.101	0.057
	*O	0.063	0.083	0.040
	*OOH	0.429	0.194	0.097
(X-sub)-Co SAs-WSe ₂	*OH	0.340	0.103	0.057
	*O	0.054	0.041	0.021
	*OOH	0.437	0.124	0.070

^a The calculated results under zero electric field were used uniformly, because electric field has negligible influence on the thermodynamic corrections.

REVIEWERS' COMMENTS

Reviewer #3 (Remarks to the Author):

I am satisfied with the authors' latest response. This manuscript should be accepted for publication in Nature Communications.

Point-by-Point response to Reviewers

Reviewer #1:

The authors have adequately addressed all reviewer comments.

Response: We greatly appreciate the reviewer's positive comments.

Reviewer #2:

I am pleased with the authors' answer to my comments and the corresponding changes brought to the manuscript. They have addressed the concerns that I raised in the initial review. In the revised version and the response to my comments, the authors have added the combination of XPS, XANES, EXAFS and STEM characterizations to confirm the precise information related to the valance state and coordination environment of single-atom metals. They also emphasize the concept of OEEF-accelerated chemical reactions at the atomic level, and highlights the novelty. Additionally, the stability of TMDs during OER has been further carefully confirmed. Therefore, it is now ready for publication.

Response: We greatly appreciate the reviewer's positive comments.

Reviewer #3:

I am satisfied with the authors' latest response. This manuscript should be accepted for publication in Nature Communications.

Response: We greatly appreciate the reviewer's positive comments.